

# Kinematics of surface currents at the northern margin of the Gulf of Cadiz

Luciano de Oliveira Júnior[1], Paulo Relvas[2], Erwan Garel[1]

[1]Centre for Marine and Environmental Research (CIMA), University of Algarve, Faro, 8005-139, Portugal
5  [2]Centre of Marine Sciences (CCMAR), University of Algarve, Faro, 8005-139, Portugal

*Correspondence to*: Luciano de Oliveira Júnior (lojunior@ualg.pt)

**Abstract.** The subtidal surface water circulation at the northern margin of the Gulf of Cadiz, at the southern extremity of the Iberian upwelling system, is described based on validated hourly high frequency radar measurements from 2016 to 2020. Statistical analyses (mean, standard deviation, eccentricity and empirical orthogonal functions) are applied to the dataset, 10  which is completed with ADCP time-series from multiple moorings at 5 inner-shelf stations and ERA5 wind. Off the shelf, the main circulation pattern consists of a slope current, best developed in summer when north-westerlies dominate, in particular at the most exposed western region. Other mechanisms than upwelling must contribute to this flow in order to explain its seasonal persistence. The slope circulation reverses for regional wind events with east component $> 10$ m.s$^{-1}$, approximately. On the shelf, currents are mainly alongshore and balanced. The circulation is generally continuous along the 15  coast, except for weak ($< 0.1$ m.s$^{-1}$, broadly) poleward flows. In the latter case, the flow tends to remain equatorward near Cape Santa Maria. In winter, coastal poleward flows often extend over the entire margin and are mainly wind-driven. In summer, these flows generally consist of coastal counter currents (CCCs) with poleward direction opposed to the one of the slope current. The CCCs are associated with significant cyclonic recirculation, strongest at West, where a transient eddy is shortly observed for weak wind stress. This circulation develops after periods of strong north-westerlies, supporting that 20  CCCs result from the unbalance of a regional along-shore pressure gradient.

## 1 Introduction

The northern margin of the Gulf of Cadiz (NMGoC), along the southwest coast of the Iberian Peninsula, is characterized by a complex water circulation related to its geographic setting. The region is bounded at west by the Portuguese branch of the Canary Current Upwelling System and, at east, by the Strait of Gibraltar where important water exchange and mixing occur 25  between Atlantic and Mediterranean waters (García-Lafuente et al., 2011; Price et al., 1993). The water circulation at the NMGoC is influenced by these remote forcings together with regional wind conditions producing coastal upwelling and associated mesoscale structures (Criado-Aldeanueva et al., 2006; García-Lafuente et al., 2006; Peliz et al., 2007; Relvas and Barton, 2002; Sánchez et al., 2007; Sánchez and Relvas, 2003). Understanding the main circulation patterns is essential to support the management of socio-economic activities and of the marine ecosystem. In particular, fisheries and coastal




tourism have a considerable weight in the region (Ortega et al., 2013) and some spots on the shelf have been recognized as biodiversity sanctuaries (Boavida et al., 2016). The offshore region is also a busy maritime route (Nunes et al., 2020) for large tankers that pose a risk regarding hazardous substances spill. However, available studies about the coastal and shelf circulation are supported by relatively few direct observations, mostly in spring and summer, and provide an incomplete description of the general circulation pattern and its seasonal variability.

The large-scale surface circulation at the NMGoC has been mainly assessed from sea surface temperature (SST) satellite imagery (Fiúza et al., 1982; Folkard et al., 1997; Relvas and Barton, 2002; Stevenson, 1977; Vargas et al., 2003) and CTD measurements (Criado-Aldeanueva et al., 2006; Garcia et al., 2002; Sánchez and Relvas, 2003). These data limit the scope of investigation to water masses having a significant temperature contrast and to geostrophic flows. In situ velocity measurements were obtained from few cross-shelf ADCP transects (Cravo et al., 2013; García-Lafuente et al., 2006; García

Lafuente and Ruiz, 2007; Relvas and Barton, 2005) and week-to-months long seabed moorings at the eastern part of the inner-shelf (Criado-Aldeanueva et al., 2009; de Oliveira Júnior et al., 2021; Garel et al., 2016; Prieto et al., 2009; Sánchez et al., 2006). Besides, numerical models have been developed to investigate the wind-driven coastal circulation (Teles-Machado et al., 2007) and the hydrodynamic effects in the region of the water exchange with the Mediterranean Sea (Kida et al., 2008; Peliz et al., 2013, 2009, 2007).

From the above studies, the subtidal inner-shelf (or coastal) circulation is generally described as being dominated by alongshore flows with opposed direction and contrasted temperature in summer, with variations up to 2°C per day (Garel et al., 2016). Cold equatorward flows (EFs, broadly eastward) are generally associated to upwelling events (Fiúza et al., 1982; Relvas and Barton, 2005, 2002) while warm poleward flows (PFs, broadly westward), often referred to as coastal counter currents (CCCs), develop when upwelling favourable winds relax or reverse (de Oliveira Júnior et al., 2021; Garel et al.,

2016; Relvas and Barton, 2002; Sánchez et al., 2006; Teles-Machado et al., 2007). Observations from ADCP moorings at the eastern inner-shelf indicate that the coastal flow is highly polarized, switching semi-weekly between equatorward and poleward without clearly predominant direction during the year (de Oliveira Júnior et al., 2021; Garel et al., 2016). Cross-shelf transects further suggest that in spring and summer the CCCs constitute the northern branches of cyclonic cells that occupy the whole margin (García-Lafuente et al., 2006). At the southern boundary of the shelf, over the shelf slope, the

upper layer circulation is dominated by a permanent strong eastward current (Criado-Aldeanueva et al., 2006; García-Lafuente et al., 2006; García Lafuente and Ruiz, 2007; Peliz et al., 2009, 2007; Relvas and Barton, 2005, 2002; Sánchez and Relvas, 2003), associated to a cold SST signal in summer which is typical of upwelling events (Fiúza, 1983; Folkard et al., 1997; Relvas and Barton, 2002; Vargas et al., 2003). This feature has been termed a "slope current", not in the sense of JEBAR driven (Simpson and Sharples, 2012) but that it is somehow constrained by the slope bathymetry (Peliz et al., 2009,

2007; Relvas and Barton, 2002; Sánchez and Relvas, 2003).

To contribute to the knowledge of the water circulation at the NMGoC, the present study addresses the kinematics of surface currents based on 4.5 years (February 2016 - October 2020) of hourly measurements from the South Iberian High Frequency Radar (HFR) system. The analysis allows the establishment of the main circulation patterns and its variability. Special



attention is paid to the distribution and seasonality of the coastal and slope flows and to their linkage through cross-shore

recirculation in relation to wind conditions. The results provide a detailed characterization of the surface circulation at the

NMGoC and some insights about their driving processes.

## 2 Study Area

### 2.1 Geographical Setting

The NMGoC lies along the southern Atlantic coast of Portugal and Spain. It extends from Cape São Vicente (CSV), where

the coastline orientation changes from meridional to zonal at the southwest of Portugal, to the Strait of Gibraltar at east

(Figure 1). The margin consists of two distinct physiographic regions separated by Cape Santa Maria (CSM) where the shelf

is the narrowest (5 km-wide): a western bight, characterized by a relatively narrow shelf (< 30 km) with a steep slope; and an

eastern bight where the shelf is comparatively wider (> 40 km) and the slope is gentler (Figure 1). The shelf break is at about

200 m in depth. The few rivers flowing into the NMGoC are mainly located at east (e.g., the Guadiana, Tinto-Odiel and

75    Guadalquivir in the study area; Figure 1) and feature a low freshwater discharge throughout the year due to the semi-arid

regional climate and to strong river flow regulation by dams (Díez-Minguito et al., 2012; Garel and D'Alimonte, 2017).

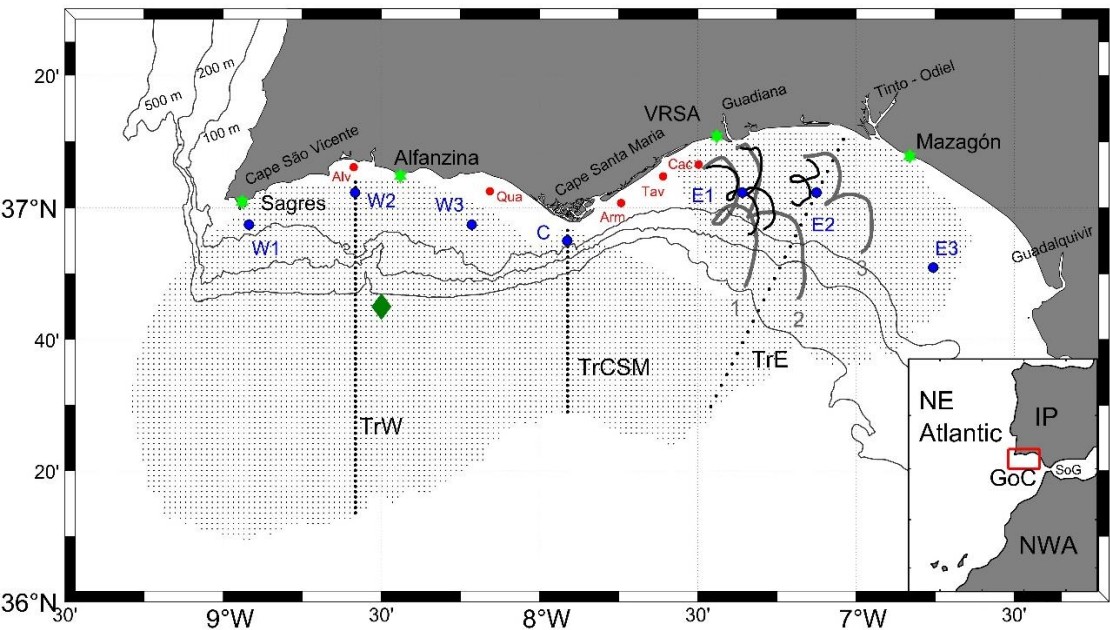

**Figure 1 Study area with location of the HFR antennas (green stars, with VRSA: Vila Real de Santo Antonio), ADCP mooring (red dots, with Alv: Alvor, Qua: Quarteira, Arm: Armona, Tav: Tavira, Cac: Cacela.), HFR grid nodes with ≥ 60% of measurements**

80    **(thin black dots) along with the transects (TrW, TrCSM and TrE indicated as thick black dots) and grid nodes (W1, W2, W3, C, E1, E2, E3, thick blue dots) analysed in the study. The thick grey and black lines represent the drifters' trajectories and corresponding PVD from HFR data, respectively (see section 4). Dark green diamond indicates the point where wind from ERA5 reanalysis was extracted (Section 6.1). The isobaths of 100, 200 and 500 m are represented as thin black lines. For general location, see inset (IP: Iberian Peninsula; NWA: Northwest of Africa; GoC: Gulf of Cadiz and SoG: Strait of Gibraltar).**





## 2.2 Circulation Patterns

Coastal upwelling generally occurs from April to September along the west coast of Portugal due to the predominance of northerlies (Alvarez et al., 2008; Fiúza et al., 1982). As the coastline sharply changes its orientation, northerlies rotate counterclockwise around CSV due to a low-pressure cell centred over the Iberian Peninsula and to orographic constraints induced by the presence of a coastal mountain range (Fiúza, 1983; Relvas and Barton, 2002). The westerly component of the rotated wind may promote coastal upwelling along the NMGoC until 7°15'W approximately, being generally more pronounced at the capes (CSV and CSM) (Criado-Aldeanueva et al., 2006; Relvas and Barton, 2002). These events generally last for few days only, such as the NMGoC has been described as a region with episodic upwelling events rather than a typical upwelling region (such as western Iberia) where upwelling persists during a substantial part of the year, at least (Garel et al., 2016).

The equatorward upwelling jet over the western Portugal shelf tends to follow the coast around CSV and to merge with locally upwelled water at the NMGoC (Relvas and Barton, 2005, 2002; Sánchez and Relvas, 2003). There, the flow typically corresponds to a band of cold SST along the shelf and its slope (Fiúza, 1983; Folkard et al., 1997; Relvas and Barton, 2005; Stevenson, 1977; Vargas et al., 2003) which eastward extension is promoted by favourable (westerly) wind (Criado-Aldeanueva et al., 2006; Vargas et al., 2003). Velocity measurements have confirmed that this cold water band is associated with eastward currents, having relatively strong near-surface velocities ($> 0.25$ m.s$^{-1}$) along the slope (Cravo et al., 2013; García-Lafuente et al., 2006; Peliz et al., 2009; Relvas and Barton, 2005). Over the western bight, this current has been observed up to 300 m in depth and to extend significantly offshore from the slope in summer (García-Lafuente et al., 2006). At CSM, the slope current approaches close to the coastline due to the narrowness of the shelf (Cravo et al., 2013; Criado-Aldeanueva et al., 2006). Over the eastern bight, the flow has been reported during all seasons and veers anticyclonically following the slope orientation (Figure 1; Criado-Aldeanueva et al., 2009, 2006; Fiúza, 1983; Garcia et al., 2002; Peliz et al., 2009, 2007; Relvas and Barton, 2002; Sánchez and Relvas, 2003). Measurements from ADCP moorings suggest that, at a sub-monthly scale, these flows reverse predominantly in winter and are wind-driven (Criado-Aldeanueva et al., 2009). In addition, numerical model results support that the Mediterranean inflow-outflow coupling contribute significantly to the development of the slope current through an entrainment process (Peliz et al., 2009, 2007). These authors proposed to name this current the Gulf of Cadiz Current (GCC).

Over the inner-shelf, the polarized alongshore subtidal circulation (de Oliveira Júnior et al., 2021; Garel et al., 2016) is well-evidenced on SST images from spring to autumn due to strong thermal contrast (Fiúza, 1983; Folkard et al., 1997; Relvas and Barton, 2002). The upwelled cold water is frequently displaced offshore by a narrow band of warm water, about 10-20 km wide, leaning along the coast. This warm water signal originates from the region of the Guadalquivir mouth (Figure 1) and propagates westward depending on the strength and duration of easterlies, rarely reaching the West coast at north of CSV (Fiúza, 1983; Relvas and Barton, 2002). The corresponding PFs (so-called CCCs) are produced by the unbalance of an alongshore pressure gradient during the relaxation (or reverse) of upwelling favourable winds (de Oliveira Júnior et al.,



2021; García-Lafuente et al., 2006; Garel et al., 2016; Relvas and Barton, 2002) and are enhanced by easterlies (Teles-Machado et al., 2007). ADCP measurements at the eastern inner-shelf show that EFs and PFs occur equally along the year,

reversing direction every 4 days in average (de Oliveira Júnior et al., 2021; Garel et al., 2016). Cross-shelf ADCP transects (García-Lafuente et al., 2006) and a spring-summer climatological analysis of the geostrophic surface circulation based on historical (1900-1998) CTD data (Sánchez and Relvas, 2003) suggest the existence of two cyclonic cells centred over the eastern and western bights, connecting the slope and coastal flows.

## 3 Data and Methods

### 3.1 HFR, ADCP and Drifter Data Sets

The study area is equipped with four CODAR medium range SeaSonde HFR antennas located in Sagres, Alfazina, Vila Real de Santo Antonio (VRSA) and Mazagon (Figure 1, green stars), as a result of a collaboration between Puertos del Estado (Spain) and Instituto Hidrográfico (Portugal). The system operates at 13.5 MHz, providing hourly radial surface velocities with spatial resolution of approximately 1.5 km up to 60 km from the coast (CMEMS Service Evolution, 2017). Each

antenna measures the velocity towards or away from it; thus, at least two antennas are required to compute the total velocities (zonal and meridional components) through least squares fitting (Lipa and Barrick, 1983; Paduan and Washburn, 2013). At regions where the radials from two antennas make an angle $\leq 20°$, the orthogonal velocity component cannot be estimated accurately (Chapman et al., 1997; Paduan and Washburn, 2013) and is estimated from adjacent valid measurements (CODAR, 2004a, 2004b).

The first pair of HFR antennas, at VRSA and Mazagon, was installed in 2013 covering an area restricted to the eastern bight. Alfazina station started operating in November 2014, extending the spatial coverage westward of CSM (up to 8°20'W). In February 2016, the last antenna was installed in Sagres, and full coverage of the western shelf was achieved (Figure 1). The dataset analysed in this study corresponds to the period with largest coverage, from February 2016 to October 2020. Earlier data were used for validation.

ADCP records were obtained at 5 moorings stations along the coast (Armona, Cacela, Tavira, Alvor and Quarteira) at water depths of 20-23 m (for location, see red stars in Figure 1). A total of 30 deployments, lasting 0.4 to 6 months each, were performed between 2008 and 2019 using Workhorse 600 kHz and Sentinel V 500 kHz ADCPs from TRDI (Figure 2). For each deployment, the instrument was installed inside a cubic concrete artificial reef unit (1.4 m side) lying on the bottom, with the sensor head slightly rising out. Velocities were recorded along the water column within cells of 0.5-1 m in thickness

(depending on the deployment) with a sampling interval of 60 minutes, at maximum. The standard deviation (STD) of the horizontal velocity resulting from the ADCP setup (number of pings per ensemble, cell size, etc.) was generally less than 0.03 m.s$^{-1}$.

Three Metocean iSPHERE drifters were deployed by Instituto Hidrográfico on 10 May 2013 at 2-8 km from the eastern bight shore, between 7°W and 7°30'W (Figure 1). The drifters weight 13.15 kg for a diameter of 34 cm. They have no





drogue, making the drift relatively sensible to wind conditions. The drifters' position was recorded every 10 minutes by an

internal GPS.

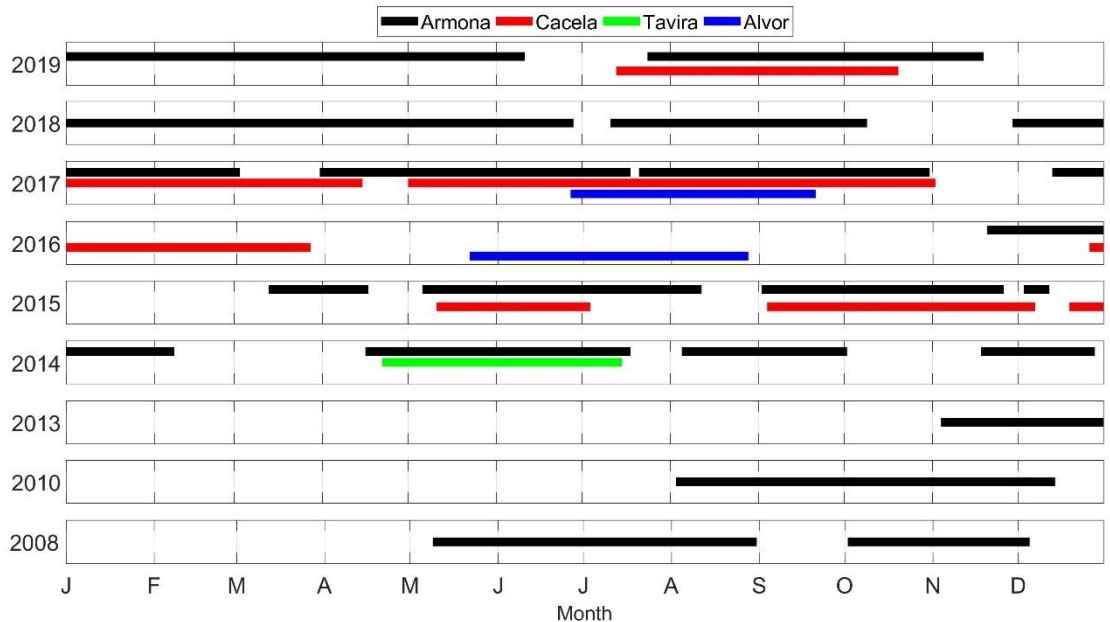

**Figure 2. ADCP deployments per month (x axis) between 2008 and 2019 (y axis) at Armona (black), Cacela (red), Tavira (green), Alvor (blue) and Quarteira (orange) stations.**

**3.2 Processing**

ADCP data quality was ensured by independent validation of each ensemble following the procedure described in Garel et al. (2016). In particular, the upper cells affected by the surface boundary were removed based on the signal intensity. Validated cells were generally from 3 m off the bed to about 80% of the water depth. Only the uppermost validated cell was considered in this study.

The HFR maps with low spatial coverage ($< 50\%$) were removed from the time-series. Subsequently, periods with infrequent consecutive maps were also discarded resulting in data gaps ranging from 2 days up to 144 days (see blanks on Figure 3a). The zonal and meridional surface velocity components were linearly interpolated at grid nodes with time gaps $\leq 6h$.

The HFR and ADCP velocity components were low-pass filtered with a Butterworth filter of 40-hour cut-off period. The resulting subtidal (or sub-inertial) zonal (u, positive eastward) and meridional (v, positive northward) velocities are

considered hereafter, unless indicated. For HFR data, the mean, STD ellipses and eccentricity ($= \frac{\sqrt{a^2-b^2}}{a}$, where $a$ and $b$ are the length of semi-major and semi-minor axis of an ellipse respectively) maps were produced for the region having at least 60% of records at each grid node (Figure 3b). This threshold allows considering a large area with few temporal gaps. For instance, the hourly velocity maps cover at least 80% of the selected area during 90% of the period 2016-2020 (Figure 3a).



The analysis was performed considering both the whole time series and seasons (defined for simplicity as winter: 1

December-28 February; spring: 1 March-31 May; summer: 1 June 31 August; and autumn: 1 September-30 November).

**Figure 3. a) Temporal distribution of the spatial coverage area considering grid nodes having at least 60% of records. b) Percentage of data at each grid node with indication of the 60% and 75% isocontours (thick black lines). The isobaths of 100, 200 and 500 m are represented as thin black lines.**

In order to describe the surface current main variability patterns, an empirical orthogonal function (EOF) analysis was

applied to the HFR data following the techniques described in Kaihatu et al. (1998) and Kundu and Allen (1976). The

current field $V(x, t)$, where $t$ is the time and $x$ is the coordinate, is expressed as a complex scalar $V(x, t) = u(x, t) + jv(x, t)$, where $j = (-1)^{0.5}$. The data set $V$ is then decomposed in terms of $k$ spatial and $k$ temporal coefficients ($\phi_k$ and $a_k$, respectively where $k$ is an integer that ranges from 1 to the total number of grid nodes):

$V = \sum_k a_k(t)\phi_k(x)$ ,                                                                                    (1)



The spatial and temporal coefficients are complex numbers and are typically represented by their amplitude and phase. Furthermore, the complex eigenfunctions $\phi_k$ can be decomposed according to the velocity components as $\phi_k = \phi_u^k + \phi_v^k$.

The EOF analysis was performed using maps having at least 75% of spatial coverage (against 60% for the mean and STD) to avoid excessive interpolation (the method requires the dataset to be free of gaps). The velocity components were interpolated
using the EOF based method presented in Beckers and Rixen (2003), which is widely used for filing gaps of satellite derived products (Alvera-Azcárate et al., 2005) and is suitable to the case of HFR data (e.g., Hernández-Carrasco et al., 2018; Kokkini et al., 2014). The technique consists in subtracting the mean values from each time series and substituting the missing values by zero. Then, an EOF analysis is applied to the demeaned matrix in order to reconstruct the time series based on EOF modes with highest variability. This procedure is performed iteratively, substituting the originally missing
values with the estimated ones. The number of iterations and the number of modes to be retained is defined based on statistical convergence (achieved through cross-validation). The final step consists in summing the mean value back to each time series which have then no gap.

The variability of the flow from the coastal to the off-shelf regions (i.e., the region offshore the 200 m isobath, hereafter) was evaluated along transects at the western bight, CSM and eastern bight (TrW, TrCSM and TrE, respectively; thick dotted lines
in Figure 1). Each transect is approximately perpendicular to the shelf break, which roughly corresponds to the coastline orientation: TrW and TrCSM are N-S, while TrE is NE-SW. The flow is represented by its alongshore (Val) and cross-shore (Vcr) components, corresponding to u and v, respectively, for TrW and TrCSM, and to u and v rotated 30° clockwise from east for TrE. The width (i.e., offshore extent from the coast) of EFs and PFs along the transects was quantified considering flows with Val of the same sign at the two most landward nodes, after smoothing out small velocity fluctuations with a 5-
nodes moving average.

The propagation of EFs and PFs along the coast was evaluated considering Val at 3 grid nodes located at a depth of 40 m (W2, C and E3 in Figure 1). At these nodes, Val was obtained based on the angle of maximum variance, which closely corresponds to the nearby coastline orientation, as previously reported at inner-shelf mooring stations (de Oliveira Júnior et al., 2021; Garel et al., 2016; Prieto et al., 2009).

**4 HFR Data Validation**

ADCP time series at Alvor, Tavira and Cacela Stations were compared with HFR velocities at the nearest grid nodes to estimate the quality of HFR data near the coast. The selected nodes were located at less than 1 km for Cacela and Tavira (which are both within the HFR coverage area) and at 4 km southward for Alvor. The other stations were not considered as Armona is well outside the HFR coverage area (see Figure 1) and Quarteira records (in 2014-2015; Figure 2) do not overlap
with HFR ones.

The mean ADCP velocity of the flow components is generally close to 0, while the STD of u is one order of magnitude larger than v, confirming that the coastal flow is mainly alongshore and polarized (Table 1). The HFR velocities feature



similar characteristics, except at Alvor grid node (where the mean of v is larger than the mean of u) possibly due to the distance between the HFR node and ADCP station. The deployment at Cacela from December 2016 to April 2017 illustrates

the good correspondence between HFR and ADCP records and the predominance of the u flow component (Figure 4). Overall, the Spearman's correlation coefficient (R) between HFR and ADCP is very good (0.92) for u and poor for the weak v component (Table 1). Furthermore, the mean of the differences and the root mean square of the differences (RMSd) between HFR and ADCP velocities are small ($\leq 0.09$ m.s$^{-1}$ and $\leq 0.11$ m.s$^{-1}$ respectively). The present skill scores are similar to those obtained at regions with flow velocities similar to the ones at the NMGoC (Lorente et al., 2015). Despite some

expected differences between HFR and ADCP velocities due to their distinct measurement methods (e.g., in term of sampling and averaging duration, horizontal position, measurement depth and footprint), the analysis supports the good quality of the HFR measurements, in particular near the coast.

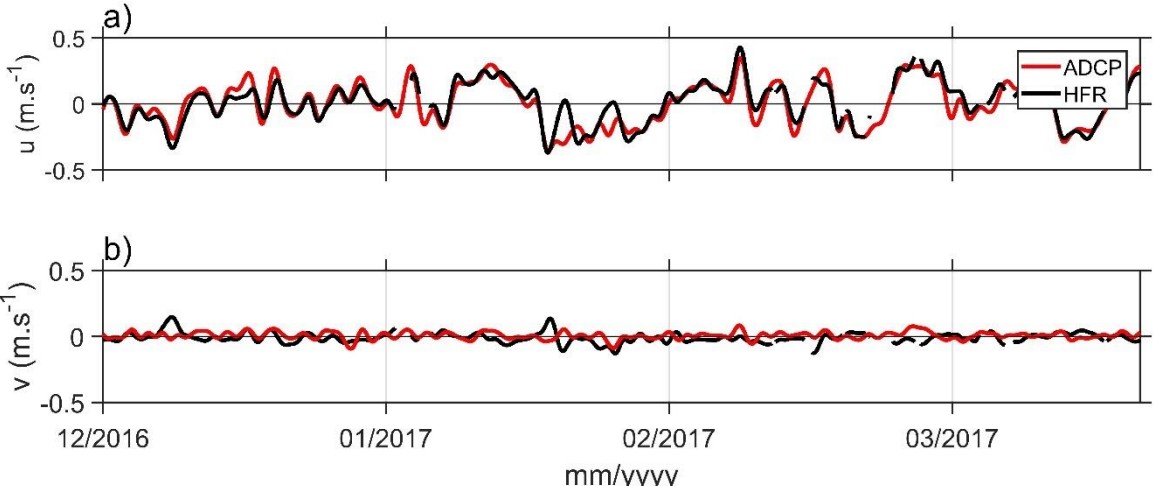

**Figure 4. Comparison of the flow velocity (m.s$^{-1}$) recorded by an ADCP at Cacela station (red lines) with the velocity at the nearest**
**HFR grid node (black lines) from December 2016 to April 2017: a) eastward component u; b) northward component v.**

On the shelf, HFR records were compared with drifters' trajectories. Drifters' velocities were obtained from their positions subsampled at the HFR sampling time. At each time step, the (unfiltered) HFR velocity at the nearest grid node to the drifter position was extracted to produce a progressive vector diagram (PVD). The trajectories of the three drifters presented a general southward displacement of 31-45 km affected by clockwise inertial rotation (Figure 1, grey lines). Although shorter

offshore (in particular for drifter 3), the drift was well reproduced by the PVD in all the 3 cases (Figure 1, black lines). The statistical correspondence between the drifters and HFR flow components is poorer than for HFR-ADCP data (Table 1). Discrepancies between HFR and drifter velocities are inherent to their (distinct) acquisition techniques (e.g., spatial averaging of eulerian records for HFR against lagrangean measurements at a point for the drifters), along with the potential wind drag effect on the emerged part of the drifters. Nevertheless, the results are within the range of what has been reported

as satisfactory in other studies comparing HFR currents with various types of drifters (Kaplan et al., 2005; Paduan and Rosenfeld, 1996; Solabarrieta et al., 2014).



**Table 1. Validation statistics between HFR and in situ (i.e., ADCP, drifter) measurements of the u and v flow components.**

| In situ observations | Period | Mean (STD) in m.s$^{-1}$ | | | | R | | Mean difference (STD) in m.s$^{-1}$ In situ – HFR | | RMSd in m.s$^{-1}$ In situ – HFR | |
|---|---|---|---|---|---|---|---|---|---|---|---|
| | | u in situ | v in situ | u HFR | v HFR | u | v | u | v | u | v |
| **ADCP Cacela** | May - Jul 2015 | 0.01 (0.16) | 0 (0.03) | 0.04 (0.12) | -0.02 (0.04) | 0.92 | 0.64 | -0.03 (0.06) | 0.02 (0.03) | 0.07 | 0.03 |
| | Sep - Dec 2015 | 0 (0.13) | 0.01 (0.03) | -0.02 (0.11) | 0 (0.03) | 0.84 | 0.33 | 0.01 (0.07) | 0.01 (0.04) | 0.07 | 0.04 |
| | Dec 2015 - Mar 2016 | 0.07 (0.16) | 0.02 (0.03) | 0.08 (0.15) | -0.01 (0.03) | 0.94 | 0.19 | -0.01 (0.06) | 0.02 (0.04) | 0.06 | 0.05 |
| | Dec 2016 - Apr 2017 | 0 (0.16) | 0.01 (0.03) | 0.01 (0.15) | -0.01 (0.04) | 0.93 | 0.04 | -0.01 (0.06) | 0.02 (0.05) | 0.07 | 0.05 |
| | May - Nov 2017 | 0.02 (0.14) | 0.01 (0.03) | 0.05 (0.12) | -0.01 (0.03) | 0.91 | 0.38 | -0.03 (0.06) | 0.02 (0.04) | 0.07 | 0.04 |
| **ADCP Alvor** | May - Aug 2016 | -0.03 (0.09) | -0.01 (0.02) | -0.06 (0.15) | -0.10 (0.06) | 0.68 | 0.14 | 0.03 (0.1) | 0.09 (0.06) | 0.11 | 0.11 |
| | Aug - Sep 2017 | -0.03 (0.09) | -0.02 (0.03) | -0.02 (0.1) | -0.06 (0.04) | 0.61 | 0.31 | -0.01 (0.08) | 0.04 (0.04) | 0.08 | 0.06 |
| **ADCP Tavira** | Apr - Jul 2014 | 0.03 (0.15) | 0.01 (0.07) | 0.08 (0.14) | 0.01 (0.09) | 0.88 | 0.31 | -0.05 (0.07) | 0.01 (0.09) | 0.09 | 0.09 |
| **Drifter 1** | May 2013 | 0.08 (0.23) | -0.23 (0.31) | 0.01 (0.21) | -0.09 (0.17) | 0.86 | 0.90 | 0.07 (0.09) | -0.13 (0.18) | 0.12 | 0.23 |
| **Drifter 2** | May 2013 | 0.12 (0.21) | -0.29 (0.24) | 0.04 (0.15) | -0.18 (0.15) | 0.87 | 0.84 | 0.08 (0.12) | -0.11 (0.14) | 0.14 | 0.17 |
| **Drifter 3** | May 2013 | 0.11 (0.24) | -0.17 (0.22) | -0.03 (0.17) | -0.07 (0.1) | 0.96 | 0.66 | 0.14 (0.11) | -0.09 (0.17) | 0.18 | 0.19 |





# 5 Results

## 5.1 Mean Circulation


**Figure 5. Mean HFR surface velocities (a) and STD ellipses and eccentricity (b) for the period February 2016 - October 2020. For clarity, the ellipses and arrows are represented every three grid nodes. The mean velocity and STD ellipses of ADCP data for the deployment periods and stations indicated in Figure 2 are shown in red. The locations of the HFR antennas are indicated with green stars.**





HFR mean velocities are broadly oriented south-eastward over the study area (Figure 5a). This direction generally corresponds to the main variability of the STD (see the elongated ellipses with northwest-southeast orientation in Figure 5b), indicating the predominance of south eastward currents through time. At off-shelf regions where the current direction varies importantly, in particular between 8°10'W and 8°20'W and between 7°15'W and 7°30'W (see rounded ellipses and areas with dark blue colours in Figure 5b), the mean currents remain south-eastward. At west of 8°45'W, mean currents are

towards the south and southwest but vary principally along the northwest-southeast direction. This region is also characterized with strong velocities (see the large STD ellipses in Figure 5b). It is noted that some rays emanating from the HFR antennas feature regions with lower eccentricity than the surrounding, suggesting a slight underestimation of one of the flow components.

The main feature revealed in the mean flow is a zonal band with strong velocities (from 0.075 m.s$^{-1}$ up to 0.15 m.s$^{-1}$),

elongating east-west across the whole study area (between 36°30'N and 36°55'N, broadly). This region of intensified mean currents (RIMC, hereafter) includes the shelf slope. At the western bight, the RIMC is broader and presents greater velocities than at the eastern bight; the south-eastward mean currents are oblique with respects to the (east-west) shelf break orientation. The mean flow at the RIMC rotates cyclonically near CSV and is aligned with the shelf slope isobaths at the eastern bight due to the predominance of along-slope currents, as indicated by the STD ellipse orientations.

A well-defined region of low eccentricity values is observed near the coast (dark red in Figure 5b), except in front of CSM. These elongated STD ellipses result from the dominance of alongshore currents. The eccentricity is close to one at coastal regions where the grid nodes and antennas are aligned (e.g., near CSV), due to an underestimation of the orthogonal velocity component (that does not challenge the observed overall predominance of alongshore flows). The mean coastal flow velocity is generally > 0.05 m.s$^{-1}$ and equatorward, being poleward only near CSV (Figure 5a). This pattern is consistent with the

mean ADCP velocities, which are all alongshore and equatorward, except at Alvor station where it is poleward (see red arrows and red ellipses in Figure 5). However, considering the u-component at the 7 selected nodes on the shelf (for location, see Figure 1), the relative occurrence of EFs and PFs is balanced, except near CSM and its western flank (Table 2). The strongest mean velocities are observed at the capes (about 0.10 m.s$^{-1}$ at CSM and 0.15 m.s$^{-1}$ at CSV) and also near 8°40'W over the mid-shelf where the flow is offshore (southward). Finally, the shelf between the Guadiana and Tinto-Odiel river

mouths is characterized by variable flow directions with balanced magnitude, resulting in the weakest mean flow at the study area (< 0.025 m.s$^{-1}$ with STD = 0.15 m.s$^{-1}$). In details, the STD ellipses are elongated alongshore near the coast and along the slope at the shelf break but feature a significant cross-shelf component in between.

**Table 2. Percentage of occurrence between the eastern (u) component of eastward and westward flows at the selected nodes.**

|  | W1 | W2 | W3 | C | E1 | E2 | E3 |
|---|---|---|---|---|---|---|---|
| Eastward | 48% | 59% | 80% | 77% | 60% | 52% | 59% |
| Westward | 52% | 41% | 20% | 23% | 40% | 48% | 41% |






## 5.2 Seasonal Variability

The overall mean current direction and STD patterns (Figure 5) remain similar for all seasons, including the coastal alongshore flow delineated by low eccentricity values (Figure 6). Seasonality is mainly observed in terms of velocity magnitude at the RIMC over the western bight (Figure 6 a-d). There, the RIMC evolves from a narrow (zonal) band with
relatively weak mean currents in winter to a wide band (extending significantly off-shelf) of strong mean velocities (up to 0.2 m.s$^{-1}$) in summer. Spring corresponds to an intermediate situation between winter and summer. In autumn, mean currents are the weakest (generally < 0.075 m.s$^{-1}$) and the RIMC is poorly expressed.

Over the shelf, the mean currents are dominantly towards the SE in winter and spring (Figure 6a, b). In summer and autumn, they describe a cyclonic pattern from CSV (south-westward) to CSM (south-eastward; Figure 6c, d). The shelf region with
strong southward velocity near 8°40'W (Figure 5) is best defined in summer. For all seasons, the largest variability on the shelf (STD > 0.2 m.s$^{-1}$) corresponds to EFs at the western flank of CSM (Figure 6e-h). By contrast, the mean currents over the east margin (including the RIMC) have a relatively constant magnitude and direction for all seasons. It is noted that the cross-shelf component is enhanced at the eastern limit of the study area.







**Figure 6. Seasonal mean HFR currents and standard deviation with eccentricity as colour maps in (a, e) winter, (b, f) spring, (c, g) summer and (d, h) autumn for the period February 2016 - October 2020. For clarity, the ellipses and arrows are represented every four grid nodes**

### 5.3 Main Circulation Patterns

Modes 1 and 2 of the complex EOF analysis account for 59% of the data variability (47% and 12%, respectively). The other
modes explain no more than 6% each. The dominant spatial pattern described by mode 1 corresponds to EFs over the inner-shelf and south-eastward flows offshore having maximum amplitude at south of CSV (Figure 7a). Exceptions to this general pattern occur at west of CSM (southward shelf flows) and from the Guadiana to the Tinto-Odiel River mouths (cyclonic





rotation of the coastal flow). Mode 1 circulation is relatively constant through time, as its phase is generally close to 0 (Figure 7c). For example, it is between -25° and 25° during 47% of the time, in particular in spring and summer (70%); at

that time, the amplitude is also the highest (as illustrated by the low-passed filtered time series in Figure 7d), denoting a more vigorous circulation than in autumn and winter. Reversals of spatial mode 1 occur during any season but are relatively rare, the phase being between 155° and 205° during 9% of the time, only (Figure 7c, d).

**Figure 7. Results of the complex EOF analyses: spatial modes 1 (a) and 2 (b); temporal modes 1 (b: amplitude, c: phase) and 2 (e:**

**amplitude, f: phase). For clarity, arrows are represented every three grid nodes. The ticks on the x-axes indicate the beginning of spring and autumn. The blue line on d and f represents the low-passed filtered time series with a cut-off period of 6 months.**

Mode 2 describes a more variable circulation, both spatially and temporally, than mode 1 (Figure 7b). Velocity amplitudes are greatest over the shelf (except for the offshore area at south of CSV, as for mode 1). At the western bight, the circulation features a cyclonic cell, about 70 km in diameter, characterized by strong PFs near the coast that recirculate offshore near

CSV to merge with the region of maximum amplitude offshore. This circulation pattern occurs mainly (64%) in summer and





in autumn, when the phase is dominantly between -25° and 25° (Figure 7e). PFs are comparatively weaker at the eastern shelf and are best observed on the outer-shelf rather than inner-shelf. Noteworthy, this flow goes around CSM, thus connecting both shelves. Mode 2 is often out of phase, being for example between 155° and 205° during 48% of the whole time series (against 19% of the time between -25° and 25°), and up to 60% in winter and spring. For approximately 30% of these "out of phase" events in winter and spring, mode 1 is in phase, such as both modes contribute to the development of strong EFs over the shelf and south-eastward flows further offshore.

## 5.4 Flow Variability

**Figure 8. Hovmöller diagram of V$_{al}$ (a, c, e) and V$_{cr}$ (b, d, f) extracted at transects TrW, TrCSM and TrE, from 03 February 2016 to 01 September 2017. Equatorward and poleward velocities are represented in blue and red, respectively; onshore and offshore velocities are represented in green and orange, respectively. Black contours indicate ±0.15 m.s$^{-1}$. The 200 m isobath is indicated as a black horizontal line. Major ticks on x axes represent the first day of the indicated month and minor ticks represent one-week interval.**

 

The Hovmöller diagrams of currents at transects TrW, TrCSM and TrE show that the alongshore component is generally

stronger than the cross-shore one (Figure 8; see Figure 1 for transects location). Both components tend also to be weakest at

TrE (Figure 8e, f), as previously observed at the eastern bight on the mean and STD maps (see Figures 5 and 6). Coastal PFs

are often restricted to the shelf, i.e., up to the 200 m isobath (indicated with black horizontal lines in Figure 8). By contrast,

EFs tend to occupy the entire transects' length, especially at TrW and TrCSM. The analysis of coastal flows' width (i.e.,

cross-shore extension from land) confirms this pattern: 60-70% of PFs extends up to the shelf break (Figure 9a, c, e), while

EFs extends dominantly up to the offshore limit of each transect (Figure 9b, d, f). Cross-shore velocities are predominantly

directed offshore (orange in Figure 8). Onshore flows (green in Figure 8) may occur at any season, the strongest events being

often associated with strong PFs (red in Figure 8) along the whole transects (such as in March-June 2017) corresponding to

periods of north-westward flows over the study area.

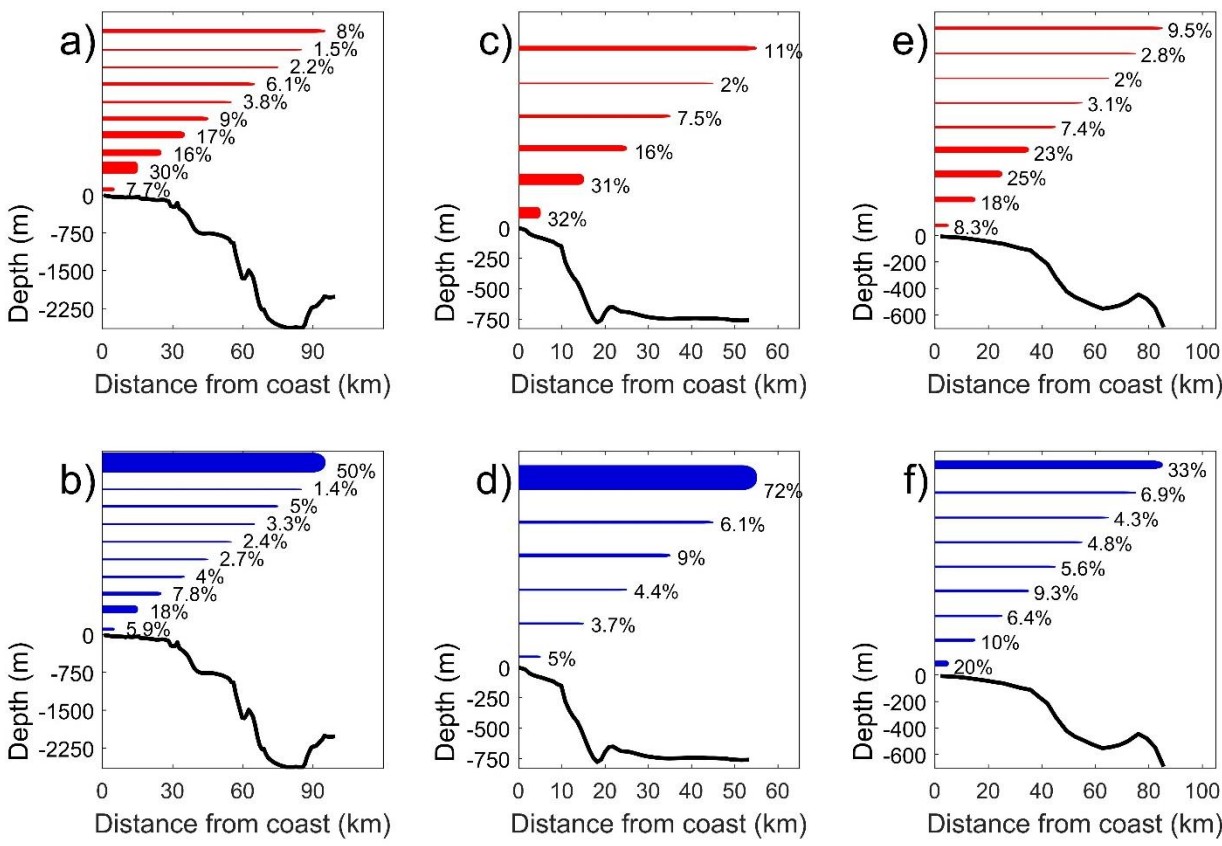

**Figure 9. Percentage of the offshore extent (from the coast) of alongshore flows (PF: red; EF: blue) at transects TrW (a-b), TrCSM (c-d) and TrE (e-f). Each bar represents the distance from the coast and bar thickness indicates the percentage. The along-transect bathymetry is represented as a black line.**

At TrW, strong offshore flows ($V_{cr}$ up to 0.5 m.s$^{-1}$) are observed over the shelf in summer and autumn (e.g., see summer

2016 in Figure 8b). During these events, $V_{al}$ is mainly poleward over the shelf and equatorward further offshore, in




agreement with the cyclonic pattern described by EOF mode 2 (also in summer and autumn) over this region (see Figure 7b, e-f). Similar observations at TrCSM also suggest an episodic cyclonic recirculation of coastal PFs in front of CSM (e.g., end of summer 2017 in Figure 8c, d).

## 6 Discussion

### 6.1 Slope Current

The present analysis of HFR subtidal currents shows that the mean surface circulation at the NMGoC is south-eastward and strongest over the slope at the so-called RIMC (Figures 5 and 6). Previous surveys have directly measured strong currents oriented along the slope with magnitude (0.15-0.2 m.s$^{-1}$) similar to the present observations (e.g., Figure 8; Cravo et al., 2013; Criado-Aldeanueva et al., 2009, 2006; García-Lafuente et al., 2006; García Lafuente and Ruiz, 2007; Peliz et al., 2009; Relvas and Barton, 2005). This current is a prominent feature of the spring-summer climatological geostrophic circulation

(Sánchez and Relvas, 2003). Numerical modelling also predicts a temporally persistent slope current, the GCC, with equivalent magnitude in the upper layer (Peliz et al., 2014, 2009, 2007).

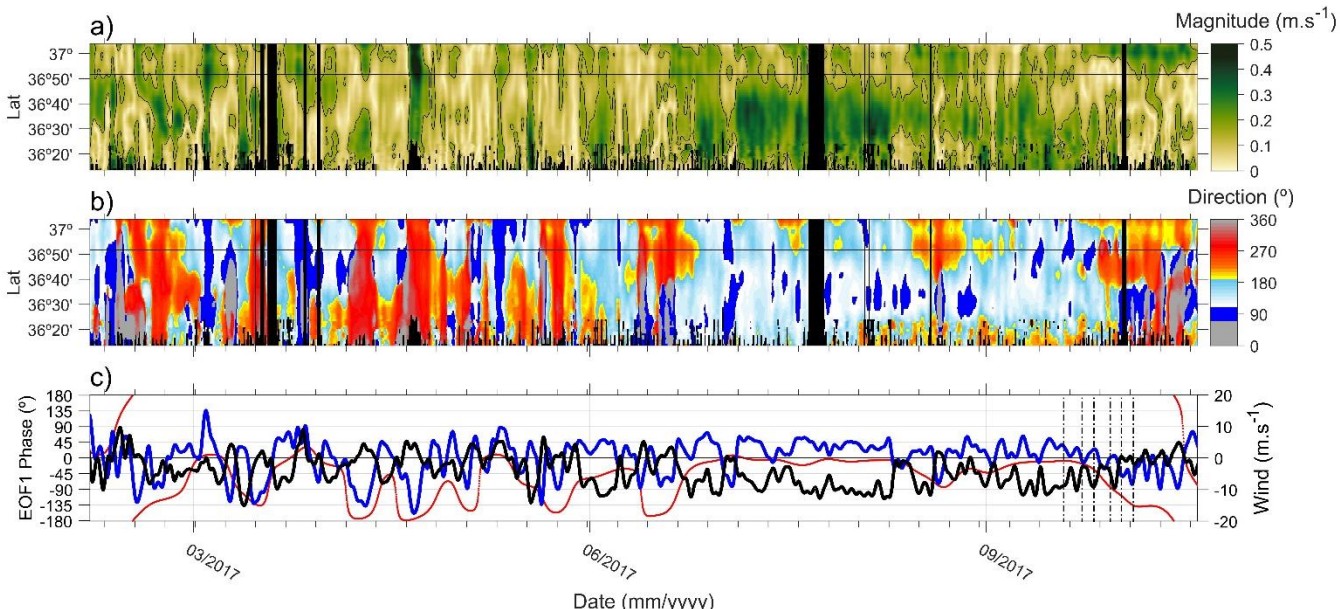

**Figure 10. Hovmöller diagram of (a) magnitude and (b) directions of currents at transect TrW; (c) zonal (blue line) and meridional (black line) of sub-inertial ERA5 wind (https://cds.climate.copernicus.eu) extracted at 8°30'W 36°45N along with the phase of EOF**
**mode 1 (red points). Dashed vertical lines indicate the day of each the SST maps represented in Figure 11. Major ticks on x axes represent the first day of the indicated month and minor ticks represent one-week interval.**

The mean HFR flow at the RIMC follows rigorously the slope at the eastern bight, in agreement with previous studies, but not at west where it is oblique to the shelf break orientation. Yet, along shelf (i.e., eastward) currents develop frequently at





the western bight, as indicated by the flow directional distribution (see the east-west elongated STD ellipses in Figure 5b). It

is noted that the off-shelf flow variability is greatest at the western border of the GoC (Figure 7a, b), the region most exposed to north-westerlies. The predominance of north-westerlies during the upwelling season (de Oliveira Júnior et al., 2021; Garel et al., 2016; Sánchez et al., 2007; Sánchez and Relvas, 2003) corresponds to a modulation of the slope circulation, which is stronger, broader and with the largest main variability in summer (Figure 6; Figure 7d, blue line). To evaluate the effect of the wind on the off-shelf circulation, the ERA5 subtidal wind at 36°45'N 8°30' (see Figure 1) is compared with the velocity

along TrW in February-October 2017 (Figure 10). In Figure 10b, the colour scale of the flow direction is designed to highlight along slope flows (i.e., eastward, in dark blue), southeast flows (light blue) and broadly westward flows (red). In winter, the slope current alternates frequently with periods of westward circulation over the entire margin associated to Levanter wind (Figure 10b, c). In summer, north-westerlies dominate (Figure 10c), and the off-shelf flow is strong (dark green in Figure 10a); the slope current (dark blue) is shifted offshore and often rotated to the southeast (light blue) along the

wind direction. It is noted that the south-eastward circulation over the western margin associated to strong north-westerlies is very similar to the mean circulation (compare Figure 5a with the example of Figure 11a which wind conditions are indicated in Figure 10c). The wind conditions that allow the development of the (eastward) along slope surface flow are not clear. However, these observations show that north-westerlies tend to deflect clockwise the surface slope current measured by HFR, as reported in other areas exposed to strong wind (e.g., Lipa et al., 2014).

At the eastern bight, CTD and SST observations suggest that the slope current constitutes the northern branch of a persistent large-scale anticyclonic cell (Sánchez and Relvas, 2003; Vargas et al., 2003). The HFR data coverage is too limited offshore to map such eddy. However, such recirculation is consistent with the strong enhancement of the cross-shelf flow component that was reported at the eastern limit of the study area (Figures 5a and 6a-d).







**Figure 11. SST from VIIRS-SNPP (https://oceandata.sci.gsfc.nasa.gov) and HFR subtidal surface currents (as arrows, which scale is indicated in Figure 11f). For clarity, arrows are represented every four grid nodes.**

The CTD and SST data indicate that the slope current has a relatively low temperature and salinity in spring-summer, typical of upwelled Atlantic waters in the GoC (Fiúza, 1983; Folkard et al., 1997; Relvas and Barton, 2002; Sánchez and Relvas, 2003; Vargas et al., 2003; see also the SST in Figure 11). Coastal upwelling produced by Ekman transport under favourable local wind is often cited as the driver of the geostrophic jet over the slope, similar to the southward jet observed along the west Iberian coast (Relvas and Barton, 2002; Sánchez and Relvas, 2003). It has also been observed that the latter southward jet turns cyclonically at CSV due to conservation of potential vorticity and progresses eastward towards the Strait of Gibraltar, merging with locally upwelled water (García-Lafuente et al., 2006; Relvas and Barton, 2002; Sánchez and Relvas, 2003). In addition, wind stress curl produced at CSV is expected to affect the water circulation at the western bight during the upwelling season (Criado-Aldeanueva et al., 2006; García-Lafuente et al., 2006; Sánchez-Leal et al., 2020; Sánchez et



al., 2007, 2006; Sánchez and Relvas, 2003), when northerlies are most frequent and intense over the west Iberian coast (Alvarez et al., 2008; Fiúza et al., 1982; Leitão et al., 2018). According to Castelao and Barth (2007), a geostrophic equatorward jet must develop offshore of the curl maxima as a response to Ekman pumping. Satellite observations in spring and summer indicate that the (monthly and seasonal) mean curl maxima may reach as south as 36°N over the western bight

(Alvarez et al., 2008; Castelao and Luo, 2018; Criado-Aldeanueva et al., 2006; Sánchez and Relvas, 2003), in agreement with the southward extent of the RIMC during these seasons (Figure 6 b-c). These processes may contribute to the development of the slope flow at the NMGoC during the upwelling season. Local upwelling in winter is also expected due to the eastward migration of the Azores high pressure cell, promoting westerlies over the GoC (Chase, 1956). Furthermore, numerical modelling simulations suggested that part of the Atlantic water is entrained by the denser Mediterranean outflow

below, producing a slope current due to mass conservation (Kida et al., 2008; Peliz et al., 2009, 2007). Since water exchange in the Strait of Gibraltar is continuous (García-lafuente et al., 2021; García-Lafuente et al., 2011), this mechanism could contribute to the observation of a slope current throughout the year, as reported in the present study.

Long-term (11 years) ADCP records at a sub monthly time-scale (i.e., low-pass filtered with a cut-off period of 40 days) over the eastern shelf slope (45 km South-eastward from TrE at 450m water depth) shows that reversals of south-eastward

flows are wind driven (Criado-Aldeanueva et al., 2009). "Levanter" wind events typically blow north-westward at the study area without clear seasonality (de Oliveira Júnior et al., 2021; Losada, 1999; Ribas-Ribas et al., 2011). As exemplified in winter 2017, the south-eastward circulation over the entire NMGoC reverses during these (strong) events (see red in Figure 10b). In addition, the circulation described by EOF mode 1 when the phase is close to 180° (see Figure 7) is remarkably associated to strong Levanters (Figure 10c), indicating the reversal of the main flow pattern over the NMGoC. These events

correspond to the ~10% of PFs occupying the entire margin on Figure 9a, c, e) and generally occur when the eastern component of ERA5 wind in the area is greater than 10 m.s$^{-1}$, approximately (not shown).

### 6.2 Shelf Circulation

The HFR and ADCP data analyses show that subtidal coastal currents are polarized in the alongshore direction at the NMGoC (Figures 4, 5, 7 and Table 2), generalizing similar findings from few ADCP mooring sites at the eastern bight (de

Oliveira Júnior et al., 2021; Garel et al., 2016; Prieto et al., 2009). EOF modes 1 and 2 indicate that the circulation is generally a regional feature, continuous along the coast (Figure 7). This coastal circulation pattern opposes the frequent disruption of PFs near CSM proposed by García-Lafuente et al. (2006) which is often cited in the literature (e.g., Casaucao et al., 2021; de Castro et al., 2017; Hanebuth et al., 2018; Mestdagh et al., 2020; Mulero-Martínez et al., 2021; Navarro et al., 2013). In agreement with García-Lafuente et al. (2006), the connection between PFs at both bights is not always clear (e.g.,

Figure 11). Comparison of the alongshore flow at various shelf locations, as exemplified in Figure 12, suggest that the PF circulation is continuous for relatively strong velocities (> 0. 1 m.s$^{-1}$). Conditioned mean maps based on the alongshore velocity at W2 show that PFs are continuous for Val > 0.1 m.s$^{-1}$ (Figure 13a, b). By contrast, EFs are always continuous (Figure 13c, d). Clearly, the setup of PFs at CSM occurs when these flows are well developed at the adjacent bights. This




delay (see 7 and 19 of April 2017 in Figure 12) explains the predominance of EFs at CSM, while EFs and PFs are balanced
elsewhere (Table 2) as previously observed at Armona Station (Garel et al., 2016). The delay is possibly due to cape-induced
bathymetric and geographic effects (e.g., Gan and Allen, 2002). In particular, the slope current is very close to the coastline
near CSM. In details, PFs from the eastern bight overshoot CSM and turn sharply northward to connect with the inner-shelf
flow at the western bight, which results in the N-S elongated STD ellipses at west of CSM (Figures 5, 6; see also the spatial
patterns of both EOF modes in Figure 7).

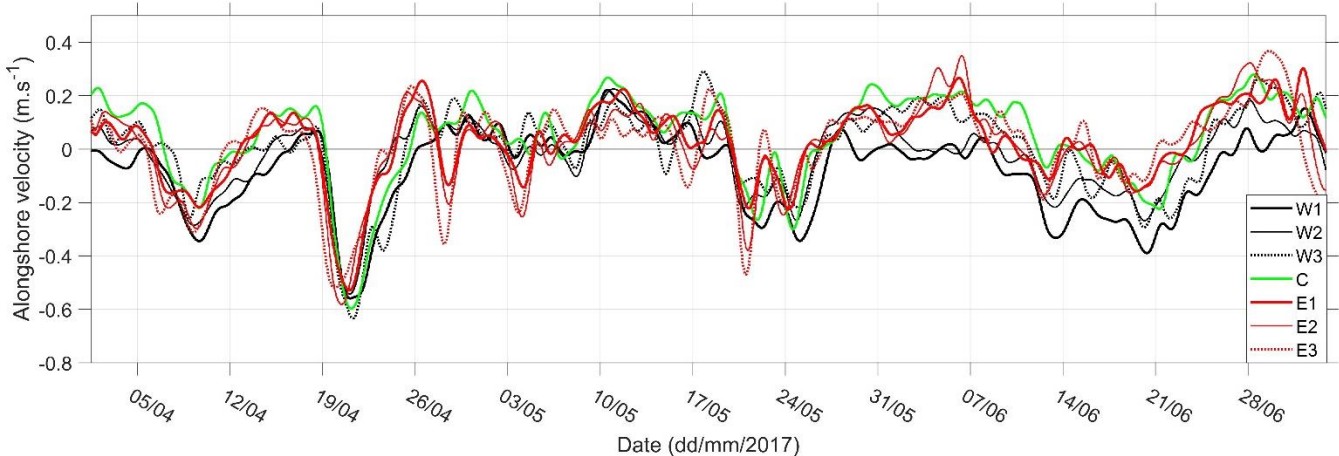


**Figure 12. Alongshore velocities at the 7 selected nodes. See Figure 1 for location. Alongshore velocities are obtained from the angle of maximum variance of velocity vectors at each node.**

Based on SST images, it has been suggested that the PF signal propagates from the eastern to the western bight (see for
example Figure 11); likewise, that EFs proceed at least partly from the west Portuguese coast (Relvas and Barton, 2002).
Such propagation patterns are not conspicuous on the subset of alongshore velocities reported in Figure 12. To evaluate
whether coastal flows develop preferentially at the eastern or western bights, the timing of EFs and PFs development is
analysed considering the 3 grid nodes W2, C, and E3 (for location, see Figure 1). Flow reversals were defined as events
occurring at the 3 grid nodes within a 7-day period. To discard small oscillations in flow direction, an event was retained
when, at each selected node, Val was $\geq 0.05$ m.s$^{-1}$ before and after reversed flows lasting 36 h, at least. A total of 23 EFs and
25 PFs reversal events were detected. In total, 61% of EFs developed first at E3 (against 17% at W2) and 48% of PFs
develop first at W2 (against 44% at E3). Sequential reversals at adjacent nodes (i.e., W2 then C then E3 for EFs; the opposite
for PFs) were defined as propagation events; no propagation event was obtained for PFs (that tend to develop latter at CSM,
as previously described) and only 3 events for EFs. Thus, coastal flows appear first at any of both bights, but tend to appear
first at the bight towards which they are directed, as illustrated on 06 June 2017 (PF developed first at west) and on 21 June
2017 (EF developed first at east) in Figure 12 (see also the early development of PFs at the western bight in Figure 11b and
c).



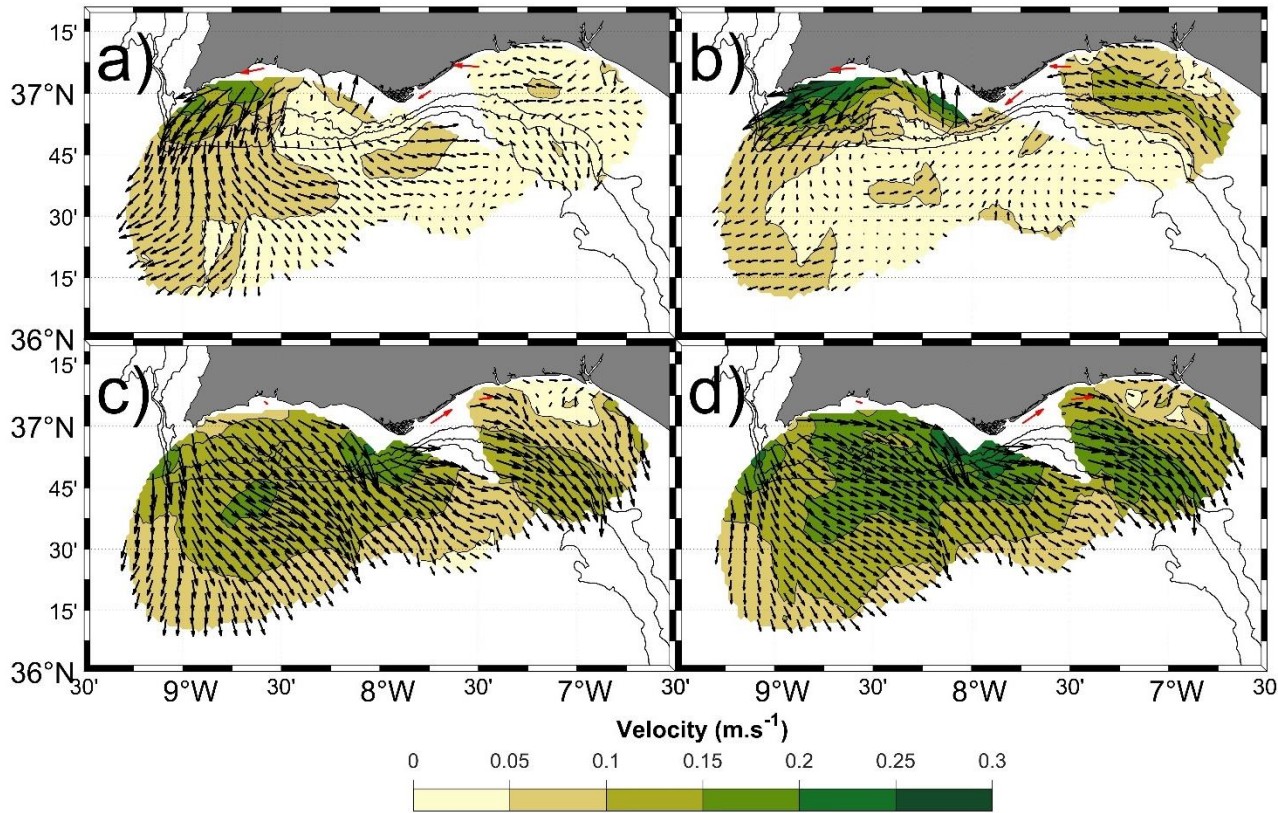

**Figure 13. Conditioned mean map computed from periods when the alongshore velocity at W2 was between 0.05 m.s$^{-1}$ and 0.1 m.s$^{-1}$ (a and c) and for periods with velocities > 0.1 m.s$^{-1}$ (b and d). Upper (lower) panel represent PFs (EFs). Red arrows indicate the mean velocity computed from available ADCP data for the same periods. For clarity, arrows are represented every four grid nodes.**

CCCs have been suggested to be driven in summer by alongshore pressure differences due strong temperature gradient between Cadiz and Huelva (García-Lafuente et al., 2006). This small-scale thermal gradient, restricted to the eastern bight, fails to explain the early setup of PFs at the western bight (where alongshore temperature variations are comparatively weaker; e.g., Vargas et al., 2003; see also Figure 11). Instead, an alongshore pressure gradient of regional scale, from the region of the Guadalquivir mouth to CSV (Relvas and Barton, 2002), is consistent with the erratic-like setup of PFs along the coast. Finally, it is noted that the 44% of PFs that started at E3 developed at W2 with an average delay of 1.24 day. This represents an average propagation speed of 2 m.s$^{-1}$ which is within the range of coastal trapped wave propagation at other systems (Maiwa et al., 2010; Rivas, 2017).

As discussed in Section 6.1, about 10% of PFs correspond to a general north-westward circulation over the entire NMGoC associated to strong Levanter wind (>10 m.s$^{-1}$). In these cases, PFs observed at the coast are mainly wind-driven and should not be considered as CCCs. For weaker wind conditions, about 60% of PFs are restricted to the shelf (Figure 9), opposed to





the dominant flow direction on the slope, and should therefore be regarded as CCCs. This spatial distribution is concordant
with SST observations of warm water near the coast and cold waters further offshore in spring and summer (Fiúza, 1983;
Folkard et al., 1997; Relvas and Barton, 2005, 2002; Reul et al., 2006) as exemplified in Figure 11. Comparisons of the flow
direction at depths of 40 m and 500 m at the transects indicate that CCCs develop predominantly (> 60%) during the
upwelling season (with maximum in late summer-early autumn) and are the rarest (< 10 %) in late autumn and winter.
Consequently, PFs in winter are mainly wind-driven while they are often CCCs (i.e., alongshore coastal flows with direction
opposed to the eastward slope current) driven by distinct processes in summer (de Oliveira Júnior et al., 2021; García-
Lafuente et al., 2006; Garel et al., 2016; Relvas and Barton, 2002; Teles-Machado et al., 2007).

## 6.3 Recirculation Between Shelf and Slope Flows

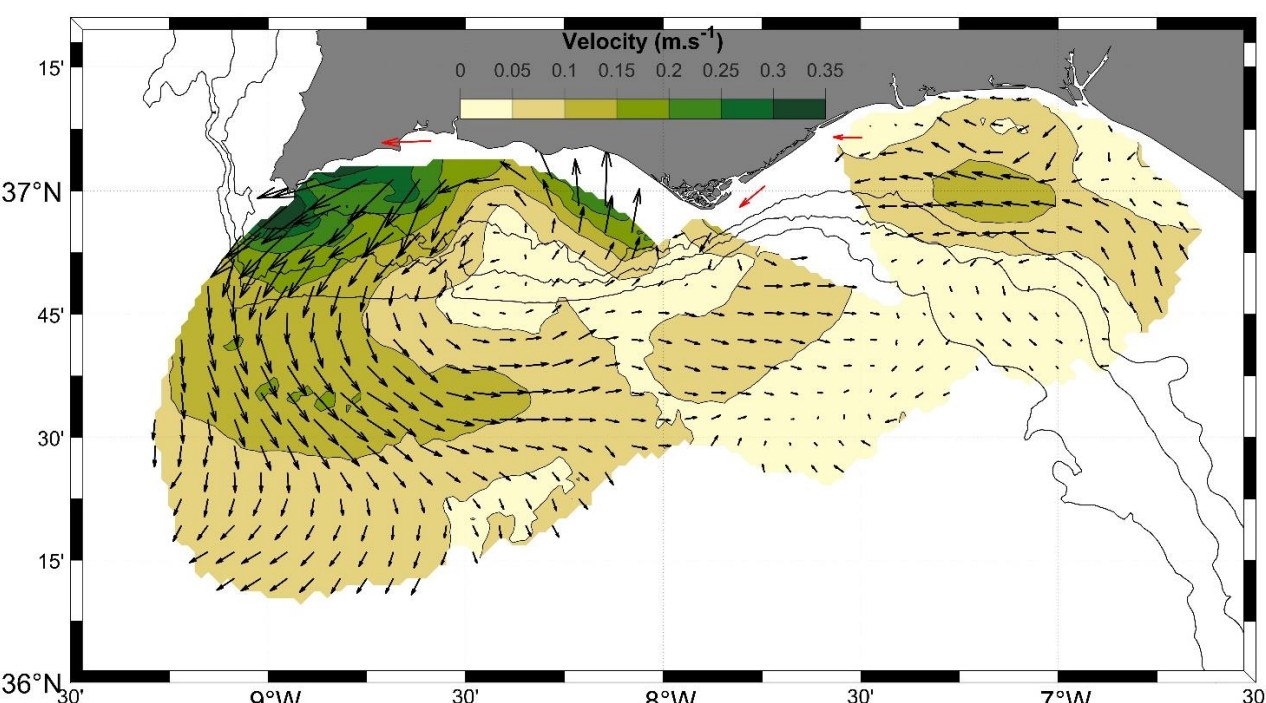

**Figure 14. Conditioned mean map computed from periods in June to October with mode 2 phase between -65° and 65° and the
ratio of mode 1 and mode 2 amplitudes ≤ 2. For clarity, arrows are represented every four grid nodes. Red arrows indicate the
mean velocity computed from available ADCP data for the same periods.**

Mode 2 of the EOF analysis (that represents 12% of the data variability) indicates episodic recirculation between the shelf
and off-shelf regions over the western bight (Figure 7b). This recirculation is cyclonic and most frequent in summer and
autumn when mode 1 is weak and mode 2 is in phase (see section 5.3). To highlight this recirculation, data are selected from
June to October when mode 2 phase is between -65° and 65° and the ratio of mode 1 and mode 2 amplitudes is ≤ 2. The
conditioned mean map obtained from these subsets outlines a cyclonic eddy over the entire western bight (Figure 14). The




northern branch of the eddy consists of a CCCs (see also the ADCP current direction in the inner-shelf, red arrows in Figure 14) that strongly recirculates offshore near CSV. This recirculation provides a mean to transport offshore coastal water-borne material such as chlorophyll (see Figure 4 in Cristina et al., 2015). It is consistent with the rare observation (based on SST) of CCCs propagating around CSV and northward along the western coast during persistent Levanter wind conditions (Relvas

and Barton, 2002). The shelf region with strong southward velocity near 8°40'W (Figure 5) which is best defined in summer results from this recirculation. The southern branch of the eddy is constituted by the slope current.

A cyclonic eddy was previously described as a quasi-permanent feature in spring and summer over the western bight (García-Lafuente et al., 2006). The positive vertical component of northerly wind curl at West of CSV produces ascending velocities resulting in an uprising of the isopycnic and advection of dense water from the ocean's interior towards the surface

(Sánchez and Relvas, 2003). Because of this upwelling process, a cyclonic circulation must develop to compensate the baroclinic pressure field (Criado-Aldeanueva et al., 2006; García-Lafuente et al., 2006). To investigate the eddy occurrence, a vector geometry–based detection algorithm was applied to the HFR time series (for details about the method, see Nencioli et al., 2010). The dataset was subsampled at each 3 grid nodes and a reduced area focused on the western bight (8°W - 9°W and 36°30'N - 37°N) was selected. Eddy centres were detected at grid points where four constraints were satisfied. These

constraints use two parameters (a and b) that can be specified in order to give flexibility to the algorithm. After several sensitivity tests, the most suitable values for a and b were defined to be 4 and 3 respectively. From the 708 detections, less than 2% occurred from November to March and more than 77% from June to October. An example is provided in August 2017 when the CCCs recirculated cyclonically after a period of general southeast flows (in Figure 15a, b). The eddy was briefly detected during the cyclonic recirculation period (blue dot in Figure 15c), followed by a period with strong offshore

shelf flows (Figure 15d). Recirculation events (identified based on the EOF criteria defined in the previous paragraph) clearly correspond to the development of CCCs, i.e., opposed shelf and slope flows (see summer-autumn 2017 in Figure 15e, where recirculation events identified by the red triangles on top). The cyclonic recirculation develops after periods of relatively strong north-westerlies (Figure 15f-g). These conditions agree with the development of CCCs during the relaxation of upwelling favourable wind supporting that they result from the unbalance of a regional along-shore pressure gradient (de

Oliveira Júnior et al., 2021; Garel et al., 2016; Relvas and Barton, 2002). The eddy is a transient feature (at least at the surface) detected during these periods, under low wind stress conditions (doted lines in Figure 15; Figure 15g; see also Figure 11).





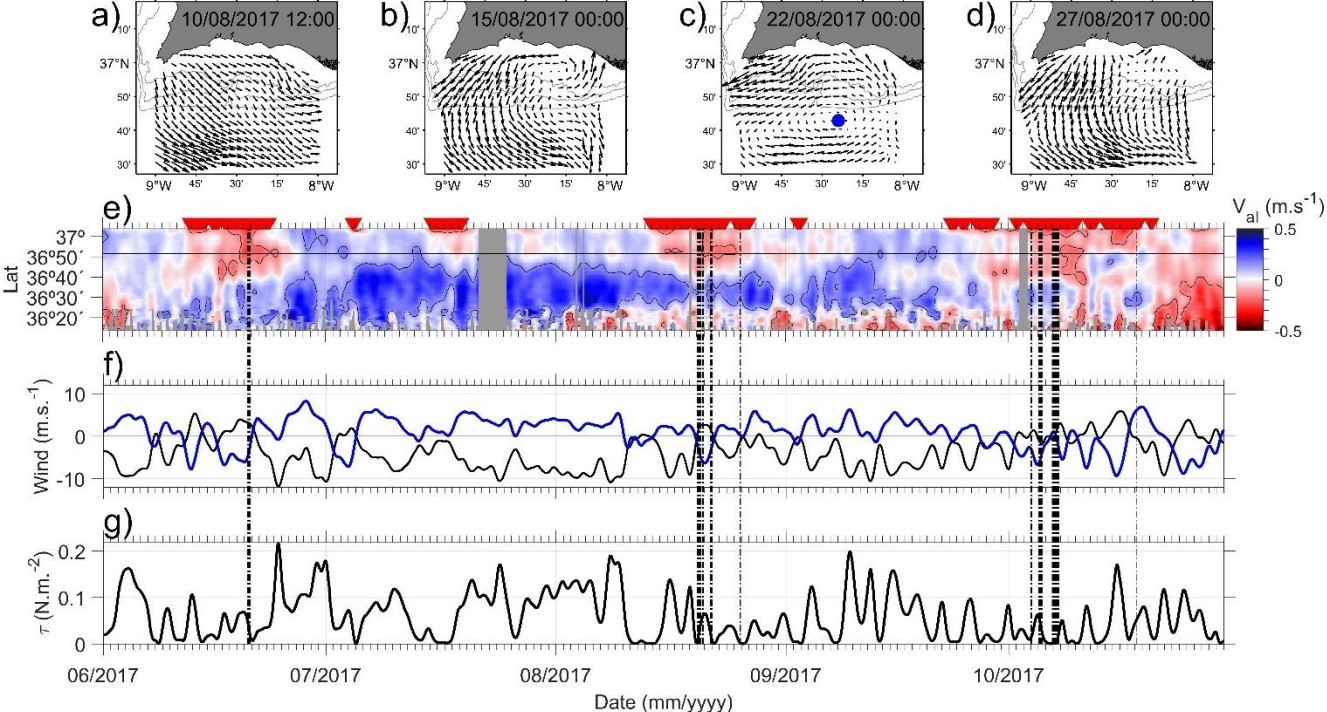

**Figure 15. Example of the cyclonic circulation evolution over the western region (a-d) and the detected cyclonic eddy centre**
**(indicated by the blue dot in c). Hovmöller diagram of filtered V$_{al}$ extracted at TrW (e). Red triangles on top represent recirculation periods identified based on EOF criteria (mode 2 phase between -65° and 65° and ratio of mode 1 and mode 2 amplitudes ≤ 2). Filtered ERA5 wind averaged at the box -9°E, -7°E, 36°45'N and 37N°, black and blue curves representing the meridional and zonal components, respectively (f). Wind stress magnitude (e). Black dotted vertical lines indicate periods when a cyclonic eddy was detected by the algorithm over the western bight. Major ticks represent the first day of the indicated month and**
**minor ticks represent one-day interval.**

García-Lafuente et al. (2006) also proposed the presence a quasi-permanent cyclonic eddy over the eastern bight. A cyclonic recirculation in this region is not apparent in the mean maps (Figures 5 and 6) and EOF analyses (Figure 7). Furthermore, the previously described algorithm yielded significantly less (60) detections at the eastern bight compared with the western bight (708). These eddies tend to develop when the western eddy is present (63%, within a time window of 36h), as exemplified in
Figure 11f. Cyclonic recirculation of the CCC was also noted at TrCSM, but more rarely than at west (compare Figures 8a-b and 8c-d). Overall, the data suggests that a cyclonic recirculation between shelf and slope flows at the eastern bight is less frequent than at west. However, it is not ruled out that this is due to the limited data coverage (see Figure 3).

## 7 Conclusions

The present study depicts the main patterns of the surface circulation at the NMGoC, based on the analysis of hourly HFR
currents from 2016 to 2020. The following conclusions are drawn, which are used to update the previous circulation sketch





of the surface circulation during the upwelling season proposed for this region for no storm conditions (Garcia-Lafuente et al., 2006). The main circulation patterns are represented as arrows which red (blue) colour indicates the direction of warm (cold) water advection (Figure 16); a wider arrow corresponds to a greater flow magnitude and a difference in the double arrowhead size represents an unbalanced flow direction; dashed arrows indicate a transient (or sporadic) circulation.

• The background circulation over the NMGoC is south-eastward as a result of the dominant north-westerlies (grey arrows in Figure 16). This circulation episodically reverses as a result of strong easterlies. Overall, the circulation is weaker at the eastern than at the western bight (as represented with the distinct arrows size in Figure 16).

• An equatorward slope current (GCC in Figure 16, following Peliz et al., 2007) is observed along the continental shelf slope, which magnitude and width is seasonally modulated (stronger and broader in summer). This flow proceeds from 535 the upwelling jet along the western coast. Strong north-westerlies tend to deflect this surface flow clockwise over the (exposed) western bight. At the eastern border of the study area, the observations support that the slope current partly recirculates anticyclonically (see Figure 16).

• Shelf flows are alongshore and balanced at the eastern and western bights (see the equal double head sizes in Figure 16), changing direction twice a week in average, without clear seasonality (Garel et al., 2016). PFs (EFs) advect warm (cold) 540 water in summer (see the blue and red arrows in Figure 16, respectively). Contrarily to the SST, the alongshore flow signal does not propagate along the coast. Instead, it tends to develop first at the bight towards which the flow is directed (i.e., PFs tend to develop first at the western shelf and EFs at the eastern shelf), consistent with a regional alongshore pressure gradient inversion.

• EFs dominate near CSM (see the distinct double arrow sizes around the cape in Figure 16) due to a delay in the 545 setup of PFs. The flow reverses when PFs are > 0.1 m.s$^{-1}$ (approximately) at the adjacent bights. Since these magnitudes are frequently reached, PFs generally go around the cape and are continuous along the coast.

• The EFs observed near the coast often extend over the entire margin as they merge offshore with the slope current (GCC).

• The PFs observed near the coast in winter are mainly associated to strong easterlies and extend over the entire 550 margin. During the upwelling season, they dominantly consist of CCCs, i.e., alongshore coastal flows with direction opposed to the equatorward slope current (Figure 16).

• At west, CCCs constitute the northern branch of a cyclonic recirculation which is strongest near CSV, promoting significant offshore transport and explaining the sporadic advection of warm water to the north of CSV (see the dashed red arrow near the cape in Figure 16). This recirculation pattern (including CCCs) develops during the relaxation of upwelling 555 favourable wind supporting that they result from the unbalance of a regional along-shore pressure gradient. For weak wind stress, a transient eddy is episodically formed, limited at south by the GCC and at east by onshore currents near CSM (see the dashed arrow near the cape in Figure 16).



•       At East, the core of alongshore flows is detached from the coast, on the outer shelf (Figure 16). Cyclonic recirculation of CCCs seems less frequent than at west (see dashed arrows in Figure 16), although this result can be due to
the limited spatial coverage of HFR data in this bight.

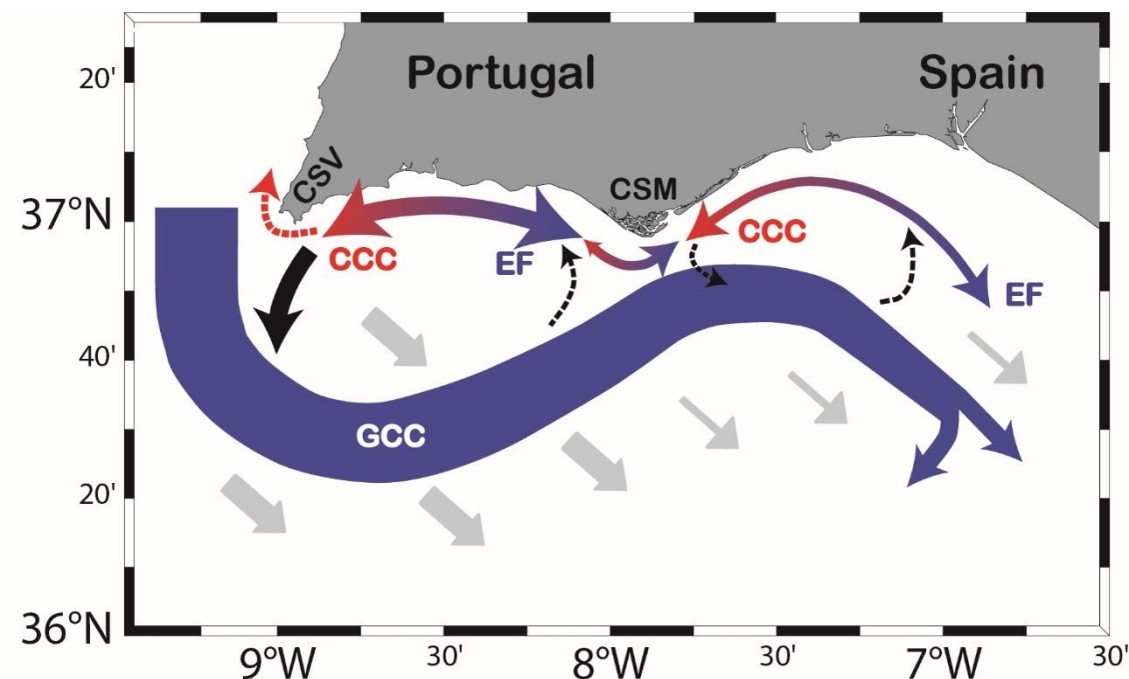

**Figure 16. Updated sketch of the main circulation patterns at the NMGoC during the upwelling season for no storm conditions.**
**The flow magnitude (schematically represented by the size of the arrows) is larger at the western bight than at the eastern bight.**
**Red (blue) arrows indicate the direction of warm (cold) water advection. Dashed arrows indicate a transient (or sporadic)**
**circulation. An equatorward slope current (the Gulf of Cadiz Current, GCC) proceeding from the West Portuguese coast and**
**advecting cold water is superimposed to the background south-eastward, wind-induced, circulation (grey arrows). The GCC partly**
**recirculates anticyclonically at east. On the shelf, the flow is alongshore and balanced between the equatorward and poleward**
**directions (as represented with equal double arrows head sizes), except near Cape Santa Maria (CSM). There, equatorward flows**
**predominate (see the distinct double arrowhead sizes around the cape) as they reverse with some delay compared with the**
**adjacent bights. However, the equatorward flows (advecting cold water) and poleward flows (advecting warm water) are generally**
**continuous along the coast, reversing twice a week, in average. Poleward flows are Coastal Counter Currents (CCCs), i.e., with**
**opposed direction than the GCC, which develop after periods of north-westerlies. At the western bight, they are associated to a**
**cyclonic recirculation, strongest near Cape São Vicente (CSV), explaining the sporadic advection of warm water to the north of the**
**cape (see the dashed red arrow near the cape). For weak wind stress, this recirculation depicts a short-lived eddy over the bight**
**due to onshore recirculation near CSM (see the dashed arrow near the cape). At the eastern bight, the CCCs and equatorward**
**flows are strongest at the outer shelf (rather than at the inner shelf at West). Cyclonic recirculation of CCCs occurs less frequently**
**than at west (see dashed arrows).**

**Data availability.**

The        HFR        observations        from        Puertos        del        Estado        can        be        downloaded        at
http://opendap.puertos.es/thredds/catalog/radar_local_huelva/catalog.html. ADCP data are available from the authors upon
request (egarel@ualg.pt).



## Author contributions

All authors contributed to the conceptualization of the study and participated on the interpretation of the results. LOJ
processed the data, plotted the results, and wrote the first version of the manuscript. EG and PR reviewed and edited the manuscript to its final version.

**Competing interests**: The authors declare that they have no conflict of interest

## Acknowledgments

HFR data are freely provided by Instituto Hidrográfico (Portugal) and Puertos del Estado (Spain) through the TRADE
project (Trans-regional radars for Environmental applications). The authors additionally thank the Instituto Hidrográfico for providing the drifter data. ERA5 wind was obtained from the European Centre for Medium-Range Weather Forecasts (ECMWF) database. The authors acknowledge the support of the Portuguese Foundation for Science and Technology (FCT) through the grant UID/MAR/00350/2020 (attributed to CIMA, University of Algarve) and grants UIDB/04326/2020, UIDP/04326/2020 and LA/P/0101/2020 (attributed to CCMAR). The work of LOJ is supported by FCT through PhD
fellowship SFRH/BD/140250/2018.

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
