# Peer review of "Kinematics of surface currents at the northern margin of the Gulf of Cadiz"

_EGUsphere, 2022_

## Referee Comment (RC1)

This paper proposes a study of the oceanic circulation in the Golf of Cadiz based on 5 years observation of surface currents (2016-2020) with 4 stations of High-Frequency Radars (HFR) deployed along the southern coast of Portugal. The combined radar stations cover an area of about 200 km along shore and 100 km offshore. The HFR currents are compared and validated with a series of in situ measurements including 3 ADCP moorings and 3 drifters.

The analysis is sound and well written. In the end it leads to a complete description of the main circulation pattern and its seasonal variations. I have no core objection to the main results and will only point out weaknesses in the methodology and possible improvements. Also, as an expert of HFR processing rather than oceanography, I will essentially comment on the techniques that are used to extract and validate the HFR surface currents.

**Main comments**

- In the Abstract it is claimed that the analysis is made from « validated hourly HFR measurements ». However, it seems that the validation and comparisons with in situ measurements is made only with low-pass filtered data corresponding roughly to a daily average (40 hours Butterworth filter). This needs some clarification. In particular, i the EOF decomposition obtained from hourly data or low-pass filtered data ? Is the interpolation of small gaps (< 6 hours) made within the EOF process or is a preliminary ad hoc filtering ? This could lead to different outcomes.</li>
- 2) As it is well known (Stewart and Joy 1974), the HFR measurement integrates the current over a depth equal to a fraction of the radar wavelength (lambda/8 pi). This make the comparison with ADCP meaningful only if the depth of the measurement cell is comparable. I could not figure out from the manuscript the exact depth of the last bin in the various ADCPs. Could this be stated explicitly and commented ? A difference of measurement depth between the ADCP could account for part of the difference of performances in the HFR comparison. This information could be given or recalled in Table 1.
- 3) Figure 4 shows the radial current on a very coarse temporal scale. It seems that during the second half of january 2017 the HFR and ADCP current have significant difference (> 20 cm/s). Is there a reason or proposed explanation for this particular period ? Could subsurface processes and current shears be responsible for this (in relation to the former point regarding the ADCP measurement depth) ?
- 4) The RMSD between HFR and drifter measurements is very large (~ 25 cm/s in norm). Due to the motion of drifter I think the comparison with daily HFR currents is not very meaningful and should rather be done with hourly data. Furthermore, the drifters having no drogue, they are more sensitive to wind and near-surface current and therefore faster than the average current over the HFR integration depth (see for example Dumas et al., Ocean Dynamics 2020 for HFR comparisons with drifters with and without drogues). All in one, the drifters do not appear to be a relevant validation tool in this context.
- 5) Did you perform self-consistency tests to assess the validity and accuracy of the EOF reconstruction? See for example Bourg & Molcard Ocean Dynamics 2021 for such kind of procedure.

Minor remarks :

- Line 133 p 5 : « ... is estimated from adjacent valid measurements ». I do not understand this statement. If the angle between radials is less than 20 degree, it will be more or less the same with adjacent measurements ?
- Line 134 p 5 : « The 2 references CODAR a,b seem to be incomplete. Are these tutorials, manuals, preprints ?
- The EOF method which is employed (Alvera 2005, Beckers and Rixen 2003) is today commonly referred to as « DINEOF ».
- At view of Fig 5 and Fig 6 on the EOF decomposition it seems than the mean field is included in Mode 1, which is 47 % of the variance. Can you please clarify whether Mode 1 in Fig 6 is a velocity anomaly or not ? Also please specify the scales and units in the plots. The amplitude of Mode 1 ranges from 0 to 20, this makes big values in the end when multiplying by the amplitudes of Mode 1 or 2.
- Regarding the phase ambiguity (180/-180) in Figure 7, this could be circumvented by plotting the cosinus or by unwrapping the phase.
- As noticed by the authors the phase of the spatial modes is close to 0 or ± 180 degree in Figure 7. Is this a criterion of correctness for the EOF decomposition ? Otherwise can one expect arbitrary values for this phase ?

---

## Author Comment (AC1)

**Response to Referee #1**

This paper proposes a study of the oceanic circulation in the Golf of Cadiz based on 5 years observation of surface currents (2016-2020) with 4 stations of High-Frequency Radars (HFR) deployed along the southern coast of Portugal. The combined radar stations cover an area of about 200 km along shore and 100 km offshore. The HFR currents are compared and validated with a series of in situ measurements including 3 ADCP moorings and 3 drifters. The analysis is sound and well written. In the end it leads to a complete description of the main circulation pattern and its seasonal variations. I have no core objection to the main results and will only point out weaknesses in the methodology and possible improvements. Also, as an expert of HFR processing rather than oceanography, I will essentially comment on the techniques that are used to extract and validate the HFR surface currents.

We are thankful for the positive evaluation of our work and have replied below (in blue) to all the Reviewer's comments. The revised text is indicated in italic, in between quotes. The line numbers correspond to the revised manuscript.

Main comments
1) In the Abstract it is claimed that the analysis is made from « validated hourly HFR measurements ». However, it seems that the validation and comparisons with in situ measurements is made only with low-pass filtered data corresponding roughly to a daily average (40 hours Butterworth filter). This needs some clarification. In particular, i the EOF decomposition obtained from hourly data or low-pass filtered data?

Reply: The original MS states in Section 4 that the validation of the HFR data is performed with filtered data (for ADCP) and non-filtered data (for drifter).

Following the Reviewer's suggestion, it is now clarified that the EOF analysis was performed with filtered data, while the DINEOF interpolation considered unfiltered data:

Lines 174-175
 "*In order to describe the surface current main variability patterns, an empirical orthogonal function (EOF) analysis was applied to the subtidal HFR data following the techniques described in Kaihatu et al. (1998) and Kundu and Allen (1976).*"

Lines 182-186
*Since EOF requires the dataset to be free of gaps, the velocity components were interpolated using the Data Interpolating Empirical Orthogonal Functions (DINEOF) method presented in Beckers and Rixen (2003), which is widely used for filing gaps of satellite derived products (Alvera-Azcárate et al., 2005) and is suitable to the case of HFR data (e.g., Hernández-Carrasco et al., 2018; Kokkini et al., 2014). The DINEOF methodology was performed using unfiltered data,*

*for maps having at least 75% of spatial coverage (against 60% for the mean and STD) to avoid excessive interpolation.* "

Is the interpolation of small gaps (< 6 hours) made within the EOF process or is a preliminary ad hoc filtering?
This could lead to different outcomes.
Reply: The interpolation is done prior to the EOF analysis. The 6h threshold was chosen to avoid excessive interpolation (generally, the flow does not change drastically during that time).

2) As it is well known (Stewart and Joy 1974), the HFR measurement integrates the current over a depth equal to a fraction of the radar wavelength (lambda/8 pi). This make the comparison with ADCP meaningful only if the depth of the measurement cell is comparable. I could not figure out from the manuscript the exact depth of the last bin in the various ADCPs. Could this be stated explicitly and commented ? A difference of measurement
depth between the ADCP could account for part of the difference of performances in the HFR comparison. This information could be given or recalled in Table 1.

Reply: We agree with the Reviewer. This point – and other causes of mismatch between HFR and ADCP velocities - was already mentioned in the original MS (Line 218-220).

Following the Reviewer advice, we have updated the MS with an indication of the ADCP near surface cell depth:

Lines 157-158
*For this study, only validated near surface cells (generally within the first 2-4 m from the surface) were considered.*

However, we choose not the ADCP cell depth in Table 1 because it is not constant.

3) Figure 4 shows the radial current on a very coarse temporal scale. It seems that during the second half of january 2017 the HFR and ADCP current have significant difference (> 20 cm/s). Is there a reason or proposed explanation for this particular period ? Could subsurface processes and current shears be responsible for this (in relation to the former point regarding the ADCP measurement depth) ?

Reply: The cause of this temporally limited mismatch is not clear. It may be attributed to any local transient surface layer phenomena, with time scale of few days, that the top bins of the ADCP did not capture In any case, such occurrence is rare and does not challenge the general good correspondence of ADCP and HFR data (as illustrated in Figure 4).

4) The RMSD between HFR and drifter measurements is very large (~ 25 cm/s in norm). Due to the motion of drifter I think the comparison with daily HFR currents is not very meaningful and should rather be done with hourly data. Furthermore, the drifters having no drogue, they are more sensitive to wind and near-surface current and therefore faster than the average current over the HFR integration depth (see for example Dumas et al., Ocean Dynamics 2020 for HFR comparisons with drifters with and without drogues). All in one, the drifters do not appear to be a relevant validation tool in this context.

Reply: Subtidal data are hourly, not daily. In any case, the analysis was conducted with the non-filtered hourly data as suggested by the Reviewer. The drifters provide the unique available dataset for comparison with HFR data offshore. Therefore, despite some limitations (and we fully agree with the Reviewer about that, as mentioned in Lines 231-233), we prefer to let this analysis in the MS. Overall, the general skill scores are within the range of values presented in the literature.

5) Did you perform self-consistency tests to assess the validity and accuracy of the EOF reconstruction ? See for example Bourg & Molcard Ocean Dynamics 2021 for such kind of procedure.

Reply: The validity and accuracy of EOF modes 1 and 2 are tested separately with HFR measurements in Figures 10, 14 and 15, providing consistent, expected correspondences:
- In Figure 10 the reversed phase ($\pm 180°$) of mode 1 coincides with broadly eastwards currents over the entire TrW (in red).
 - In Figure 14 the conditioned mean map is computed from periods with mode 2 phase between -65° and 65° and the resulting map coincides with the (in phase) mode 2 spatial map.
- In Figure 15, cyclonic eddies are only detected when mode 2 is in phase and with significant amplitude.

Minor remarks :

• Line 133 p 5: « ...is estimated from adjacent valid measurements ». I do not understand this statement. If the angle between radials is less than 20 degree, it will be more or less the same with adjacent measurements ?
Reply: The valid measurements nearby used for interpolation have an angle > 20º, as now indicated in the revised MS:

Lines 132-134
*"At regions where the radials from two antennas make an angle ≤ 20°, the orthogonal velocity component cannot be estimated accurately (Chapman et al., 1997; Paduan and Washburn, 2013) and is estimated from adjacent valid measurements (i.e., with radial angle > 20º; CODAR, 2004a, 2004b)."*

• Line 134 p 5: « The 2 references CODAR a,b seem to be incomplete. Are these tutorials, manuals, preprints?

Reply: We are thankful to the Reviewer for noticing this. The references refer to CODAR's software manual and were updated accordingly.

*"Codar: About Baseline Interpolation, Manual 2004a. http://support.codar.com/Technicians_Information_Page_for_SeaSondes/Docs/Informative/Baseline_Interpolation.pdf."*

*"Codar: Obtaining Total Current Velocities from Radials, Manual 2004b: http://support.codar.com/Technicians_Information_Page_for_SeaSondes/Docs/Informative/Combining_Radials.pdf."*

• The EOF method which is employed (Alvera 2005, Beckers and Rixen 2003) is today commonly referred to as « DINEOF ».

Reply: We are grateful for this comment. Following the reviewer's suggestion, we have included the commonly used name in the revised MS.
See Lines 183 and 185.

• At view of Fig 5 and Fig 6 on the EOF decomposition it seems than the mean field is included in Mode 1, which is 47 % of the variance. Can you please clarify whether Mode 1 in Fig 6 is a velocity anomaly or not? Also please specify the scales and units in the plots. The amplitude of Mode 1 ranges from 0 to 20, this makes big values in the end when multiplying by the amplitudes of Mode 1 or 2.

Reply: No spatial or temporal mean is removed for the EOF analyses, similar to the references cited in the text. Following the Reviewer's suggestions, the units are now indicated in Figure 7 (see below) and are explained with more detail in the revised caption.

[Figure]

Lines 304-305
*"The reconstructed velocity for each mode corresponds to the local spatial value multiplied by the dimensionless amplitude and rotated of the respective phase angle."*

The high amplitude values are counterbalanced by the small values on the spatial maps: for example, the maximum value of Mode 1 spatial map is 0.022 m.s$^{-1}$, which gives about 0.5 m.s$^{-1}$ when multiplied by the maximum temporal amplitude (23.25). This value corresponds well to the observed max value (0.6 m.s$^{-1}$) in Figure 12.

• Regarding the phase ambiguity (180/-180) in Figure 7, this could be circumvented by plotting the cosinus or by unwrapping the phase.

Reply: We are thankful for the recommendation, but after experimenting (see figure below) we believe that the original figure shows better the patterns we point out (In particular the periods when the phase oscillates around 0).

[Figure]

• As noticed by the authors the phase of the spatial modes is close to 0 or ± 180 degree in Figure 7. Is this a criterion of correctness for the EOF decomposition? Otherwise can one expect arbitrary values for this phase?

Reply: The reference to phase = 0 or ± 180 is not a criterion for the correctness of the method. It is used to indicate that the pattern represented by the spatial mode is recurrent as well as its complete reversal (when the phase is 180).

---

## Author Comment (AC2)

**Response to Referee #2**

Main comment:
The description of the general circulation of the northern area of the Gulf of Cadiz has been described based on data from the cross-border radar network in Spain and Portugal. The study is well founded, various methodologies have been applied and the analysis of the results reaches conclusions consistent with the literature. I strongly recommend the publication of the paper.

Reply: We very much appreciate the positive evaluation of our work and have replied to all the Reviewer's comments below (in blue). The revised text is indicated in italic and in between quotes. The line numbers correspond to the revised manuscript.

Minor comments:
- The surface current velocity is not the same for drifters, ADCPs and HF radars. Considering also that the surface boundary layer varies greatly in the first meters, which water parcel is measured by each instrument must be described, as well as an explanation of the possible discrepancies between one method of measurement and another.

Reply: We agree with the Reviewer. This point – and other causes of mismatch between HFR and ADCP or drifter velocities - was already mentioned in the original MS (Lines 218-220 and 231-233).
For clarity, the general depth of the near surface ADCP velocity is now indicated

Lines 157-158
"*For this study, only validated near surface cells (generally within the first 2-4 m from the surface) were considered.*"

- The drifters measurements do not provide much information to the study even in the validation phase. I would advise removing them from it.

Reply: We believe that this analysis should be included in the MS as the drifter dataset is the only one available offshore for comparison with HFR data. The general skill scores are within the range of values presented in the literature and support the good quality of HFR data.

- The text alludes to the low eccentricity of the STD ellipses when the radars are aligned. However, very close to the coast, the high eccentricity is recovered despite continuing with a high GDOP. Any explanation?
Reply: High eccentricity values (about 1) are observed when the antennas are aligned (L 260-262). See also the reply to next comment.

Typos:

Line 260, '... of low eccentricity' must say 'high eccentricity'.

Reply: Corrected (L 259). We are thankful to the Reviewer for spotting this typo and associated confusion about high/low eccentricity (e.g. previous comment).

Figure 7, caption: '...temporal modes 1 (b: amplitude, c: phase)' must say '(c: amplitude,
d: phase)'.

Reply: Corrected (Lines 303-304)

---

## Author Response (AR2)

Dear Luciano and co-authors,

Thank you for this revised version of your manuscript and the detailed response that you provided to both referees. I think the paper is in a very good shape, and almost ready. However, my advice is that the paper needs some additional minor changes before publication.

My main concern is that several of the points raised by the reviewers (and perfectly addressed by your comments) are very appropriate and raise some additional discussion points (mostly concerning the methods and validation) that should be incorporated into final manuscript. For instance:

We appreciate the positive evaluation of our work, and we are very thankful to the editor for the suggestions. We have replied below (in blue) to all comments. The revised text is indicated in italic and in between quotes. The line numbers correspond to the revised manuscript.

- Provide additional discussion in the MS on the methodological choices made for EOF computation (the previous filtering / interpolation) and the potential consequences in the results, in agreement to the comments of the reviewers and your answers.

Reply: We have followed the editor's suggestion and have commented about the similarity of the results using distinct filtering and interpolation schemes prior to the EOF computation.

Line 161 - 163

*"The zonal and meridional surface velocity components were linearly interpolated at grid nodes with time gaps ≤ 6h. This threshold assures that no excessive interpolation is performed (as the flow generally does not change drastically during such time interval). It was checked that other interpolation choices do not affect the results."*

Line 325-326

*"It is noted that the EOF results remain similar with unfiltered data. In particular, the spatial patterns of modes 1 and 2 are analogous to those obtained from filtered data and the explained variability is 42% and 9.4%, respectively."*

- Provide additional discussion in the MS on the possible impact of the depth of the first ADCP cells on the observed differences (at 13.5 MHz your radar measurements are very shallow (<0.5m), so differences can be expected when comparing with pointwise measurements at 2-4 m depth)

Reply: We have followed the editor's suggestion and have included some details about the depth measurement of both equipment.

Lines 219-225

*"Large differences up to 0.3 m.s$^{-1}$ (Figure 4a) are episodically observed. Such differences are expected due to the distinct depth of HFR and ADCP measurements. ADCPs upper measurements are at 2-4 m below the surface, while the radars measure the surface layer (< 0.5 m below surface) which is more likely affected by wind drag. Moreover, HFR and ADCP systems have distinct measurement methods (e.g., in terms of horizontal position, footprint, sampling duration and averaging). Despite these inherent differences between both equipment, the correlations between HFR and ADCP velocities support the good quality of the HFR measurements, in particular near the coast."*

- Even if not clear explanation can be given, additional discussion should be provided for the mismatch observed in January 2017 between HFR and ADCP

Reply: We have followed the editor's suggestion and have commented on the observed mismatch between the HFR and ADCP (see previous reply).

Both reviewers agree (and I do too) that the comparison with un-drogued drifters is not meaningful for validation purposes. I see you want to keep this comparison in the MS because is the only data offshore, I think you an keep it but please provide a good rationale for it – i.e. explaining this is a comparison showing a qualitative agreement between surface currents provided by the radar (surface current) and the surface drifters (surface current+wind) and consider avoiding the use of validation statistics (table 1). Did you consider performing Lagrangian comparisons? Another possibility is to check the wind in the period of comparisons (you could indeed perform quantitative comparisons with data only for periods of very low winds).

Reply: We much appreciate this comment and agree that drifters' trajectories should be compared qualitatively, only, with the PVDs. Though, we believe that there is a misunderstanding about the methods used to build the statistic table, due to our inaccurate explanations in the original MS. We used pseudo-Eulerian velocities computed from each pairs of successive drifters positions, then compared with the nearest HFR grid node velocity. This is now explicit in the revised version. If still required by the editor, we can alternatively exclude the statistics in Table 1 and keep only the qualitative comparisons of trajectories.

Lines 230-240

*"On the shelf, drifters' trajectories were qualitatively compared with HFR trajectories obtained from a progressive vector diagram (PVD) of unfiltered velocities. For statistical comparisons with unfiltered HFR data at the nearest node, drifter's pseudo-Eulerian velocities were derived from the distance between pairs of successive drifters positions, subsampled at the HFR time, divided by the time interval (1 hour).*

*The trajectories of the three drifters presented a general southward displacement of 31-45 km affected by clockwise inertial rotation (Figure 1, grey lines). Such overall drift was fairly reproduced by the PVDs in all the 3 cases (Figure 1, black lines), although they remained closer to the shore than the drifters (in particular when compared with drifter 3). The skill scores between the drifter-derived and HFR flow components is poorer than for HFR-ADCP data (Table 1). Discrepancies between HFR and drifter pseudo-Eulerian velocities are inherent to their distinct acquisition techniques (e.g., spatial averaging of eulerian records for HFR against lagrangian measurements at a point for the drifters and subsequent transformation to pseudo-Eulerian velocities), along with the potential wind drag effect on the emerged part of the drifters."*

- a typo in Line 232 "lagrangean" --> "Larangian"

Reply: Corrected (Line 239). We are thankful to the Reviewer for spotting this typo.